# LEAVE IT TO THE SPECIALIST: REPAIR SPARSE LLMS WITH SPARSE FINE-TUNING VIA SPARSITY EVOLUTION

## ABSTRACT

Large language models (LLMs) have achieved remarkable success across various tasks but face deployment challenges due to their massive computational demands. While post-training pruning methods like SparseGPT and Wanda can effectively reduce the model size, but struggle to maintain model performance at high sparsity levels, limiting their utility for downstream tasks. Existing fine-tuning methods, such as full fine-tuning and LoRA, fail to preserve sparsity as they require updating the whole dense matrices, not well-suited for sparse LLMs. In this paper, we propose **Sparsity Evolution Fine-Tuning (SEFT)**, a novel method designed specifically for sparse LLMs. SEFT dynamically evolves the sparse topology of pruned models during fine-tuning, while preserving the overall sparsity throughout the process. The strengths of SEFT lie in its ability to perform task-specific adaptation through a weight drop-and-grow strategy, enabling the pruned model to self-adapt its sparse connectivity pattern based on the target dataset. Furthermore, a sensitivity-driven pruning criterion is employed to ensure that the desired sparsity level is consistently maintained throughout fine-tuning. Our experiments on various LLMs, including LLaMA families, DeepSeek, and Mistral, across a diverse set of benchmarks demonstrate that SEFT achieves stronger performance while offering superior memory and computation efficiency compared to existing baselines. The code is provided in the supplementary material and will be released publicly.

## 1 INTRODUCTION

Large language models (LLMs) have recently demonstrated remarkable success in a wide range of complex tasks, showcasing capabilities that span language understanding (Zhu et al., 2023), reasoning (Wei et al., 2022), and code generation (Vaithilingam et al., 2022), etc. However, these models, characterized by their immense number of parameters, pose significant challenges for real-world deployment due to their computational and memory demands.

To address these challenges, researchers have proposed post-training pruning approaches to reduce the number of parameters in large language models (LLMs) without incurring the high costs of retraining from scratch (Ma et al., 2023; Zhu et al., 2024). For example, SparseGPT (Frantar & Alistarh, 2023) minimizes the discrepancy between pruned and original dense models using a layer-wise Hessian method. Furthermore, Wanda (Sun et al., 2023) introduces a metric combining output activations with weight magnitudes to identify unimportant parameters. However, post-hoc pruning methods often fail to maintain the performance, especially under high sparsity levels (e.g., $\geq 60\%$) (Lu et al., 2024a). This performance degradation poses a significant challenge to their practical application, despite the fact that pruning can substantially improve efficiency.

To maintain efficiency while preserving accuracy, a natural remedy is to fine-tune the pruned model using parameter-efficient methods (Yin et al., 2023; Lu et al., 2024c; Munoz et al., 2024). Parameter-efficient fine-tuning (PEFT) has been widely adopted to fine-tune LLMs in a computationally efficient manner (Mangrulkar et al., 2022; Pfeiffer et al., 2023; Han et al., 2024). Methods such as LoRA and its variants (Hu et al., 2021; Xu et al., 2023; Dettmers et al., 2024) significantly reduce training costs and memory footprint compared to full fine-tuning. Nevertheless, when directly applied to fine-tune sparse LLMs, these methods either incur additional storage overhead to retain the fine-tuned parameters or result in dense models after merging, which conflict with the sparse model structure and undermines the efficiency benefits of sparsity, as illustrated in Figure 1 (a).

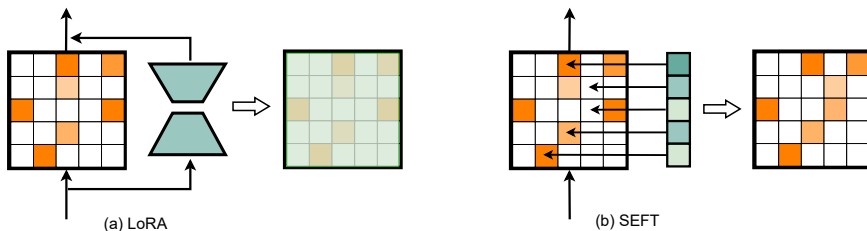

Figure 1: Comparison of LoRA and SEFT fine-tuning for sparse LLMs. While LoRA-based methods require low-rank matrices, SEFT utilizes indices (arrows) and corresponding deltas (green squares) to update LLM parameters.

More recently, to preserve the efficiency advantages of sparsity, Sparsity-Preserved Parameter-Efficient Fine-Tuning methods have been proposed (Lu et al., 2024c; Munoz et al., 2024; Hu et al., 2025). These approaches maintain sparsity after fine-tuning by incorporating specially designed adaptation mechanisms, but they often introduce additional memory and computation overhead. Moreover, these methods typically assume that the sparse topology produced by post-training pruning remains constant across the training of downstream tasks. Given the heavy reliance on calibration data in state-of-the-art LLM pruning methods, the pruning process may inadvertently remove connections that are crucial for specific downstream tasks, leading to suboptimal performance.

This raises a fundamental question: *Can we efficiently fine-tune sparse LLMs while dynamically evolving their sparse topology to better adapt to downstream tasks?*

To answer this question, we propose **Sparsity Evolution Fine-Tuning (SEFT)**, which enables the dynamic and efficient evolution of sparse connectivity during fine-tuning. This allows sparse LLMs recover task-relevant connections that may have been pruned and adapt their sparse topology throughout fine-tuning, as illustrated in Figure 1 (b).

Unlike LoRA-based approaches that introduce additional adapters, SEFT directly updates LLM parameters using a set of selected indices and corresponding update values. During fine-tuning, SEFT operates in two distinct phases: *sparse topology evolution* and *sparsity adaptation*. In the sparse topology evolution phase, parameters with small updates are dropped, while new connections are grown based on gradient information. Importantly, this allows previously inactive weights to be reactivated, enabling the sparse topology to dynamically evolve and better align with the target task. Following this, the sparsity adaptation phase restores the model to the desired sparsity level using a defined pruning criterion. This step ensures that the model remains within the predefined sparsity budget by removing excess connections introduced during the topology evolution phase, thereby maintaining both efficiency and task performance.

We evaluate SEFT on a variety of sparse LLMs from the LLaMA (Touvron et al., 2023a;b; Dubey et al., 2024) families, as well as DeepSeek (Bi et al., 2024) and Mistral (Jiang et al., 2023), with sparsity introduced via two state-of-the-art post-training pruning methods: SparseGPT and Wanda. Our experiments span models of different sizes, including 7B, 8B, and 13B, and demonstrate the effectiveness of SEFT across a diverse set of fine-tuning benchmarks, including commonsense reasoning tasks, MMLU, and GSM8K. Our contributions in this work can be summarized as follows:

- We propose SEFT, an efficient fine-tuning approach tailored to sparse LLMs, which dynamically evolves the sparse topology to enable task-specific adaptation and improve downstream performance.

- We introduce a sparsity adaptation mechanism that maintains the target sparsity level without degrading performance after topology evolution.

- We demonstrate up to a $2\times$ reduction in memory footprint and a $2.5\times$ speedup at 70% sparsity, highlighting that SEFT substantially reduces resource consumption.

- We conduct comprehensive experiments across multiple sparse LLM architectures (LLaMA, DeepSeek, and Mistral) and tasks (commonsense reasoning, MMLU, and GSM8K), validating SEFT's generality and effectiveness in both general-purpose and task-specific settings.

## 2 BACKGROUND

### 2.1 PARAMETER-EFFICIENT FINE-TUNING (PEFT)

Parameter-efficient fine-tuning (PEFT) has emerged as a popular approach to adapt LLM to downstream tasks, while significantly reducing the computational and memory overhead associated with traditional dense full fine-tuning (Pfeiffer et al., 2023; Ding et al., 2023; Han et al., 2024). Instead of updating all parameters of a pre-trained model, PEFT methods introduce a small number of trainable parameters while keeping most of the base model frozen, significantly reducing the computational overhead during fine-tuning.

One widely used PEFT method is Low-Rank Adaptation (LoRA) (Hu et al., 2021). In LoRA, low-rank matrices are introduced into specific layers (e.g., attention or feedforward layers) to capture task-specific adaptations. Specifically, let $\mathbf{W} \in \mathbb{R}^{d \times d}$ represent a frozen weight matrix in a pre-trained model. LoRA introduces a low-rank decomposition:

$$\mathbf{W}' = \mathbf{W} + \Delta\mathbf{W}, \quad \Delta\mathbf{W} = \mathbf{AB} \tag{1}$$

where $\mathbf{A} \in \mathbb{R}^{d \times r}$ and $\mathbf{B} \in \mathbb{R}^{r \times d}$ are low-rank matrices ($r \ll d$), and only $\mathbf{A}$ and $\mathbf{B}$ are trainable.

While PEFT methods like LoRA have proven effective for dense models, they are inherently not well-suited for sparse LLMs, as they either require dense updates or fail to preserve sparsity after fine-tuning.

### 2.2 SPARSE FINE-TUNING (SFT)

Unlike LoRA-based approaches, Sparse Fine-Tuning (SFT) (Guo et al., 2021; Xu et al., 2021; Ansell et al., 2022) directly modifies the base model by introducing task-specific updates in the form of a sparse "delta" vector $\boldsymbol{\delta} \in \mathbb{R}^{d_\theta}$, which is added to pre-trained parameters $\boldsymbol{\theta} \in \mathbb{R}^{d_\theta}$. The resulting fine-tuned model can be expressed as:

$$f'(\cdot; \boldsymbol{\theta}) = f(\cdot; \boldsymbol{\theta} + \boldsymbol{\delta}), \tag{2}$$

Here, $\boldsymbol{\delta}$ contains non-zero updates only at specific locations in $\boldsymbol{\theta}$, allowing for a sparse parameterization that enables efficient fine-tuning in practice.

**Sparse Parameterization.** A sparse delta vector $\boldsymbol{\delta}$ with $d_\phi$ non-zero entries can be efficiently represented using two components:

- Indices of non-zero values ($\boldsymbol{\eta}$): A vector of unique indices $\boldsymbol{\eta} \in \{1, 2, \ldots, d_\theta\}^{d_\phi}$, representing the positions within $\boldsymbol{\theta}$ where updates are applied.
- Update values ($\boldsymbol{\phi}$): The corresponding update values at these positions, denoted as $\boldsymbol{\phi} \in \mathbb{R}^{d_\phi}$.

Here, $d_\phi$ is typically a user-defined hyperparameter that controls the number of parameters updated during fine-tuning. Since $\boldsymbol{\delta}$ modifies only a small subset of the full parameter vector $\boldsymbol{\theta}$, we have $d_\phi \ll d_\theta$. This sparsity ensures that the fine-tuning process is computationally efficient and lightweight.

**Optimization Objective.** Sparse fine-tuning can be formulated as an optimization problem that aims to identify the optimal indices $\boldsymbol{\eta}$ and corresponding update values $\boldsymbol{\phi}$ to improve model performance on a given task-specific dataset $\mathcal{D}$. The objective is defined as:

$$\boldsymbol{\eta}^\star, \boldsymbol{\phi}^\star = \underset{\boldsymbol{\eta}, \boldsymbol{\phi}}{\arg\min} \mathcal{L}(\mathcal{D}; \boldsymbol{\theta}, \boldsymbol{\eta}, \boldsymbol{\phi}), \tag{3}$$

where $\mathcal{L}$ denotes the loss function that measures the discrepancy between the model predictions and the ground-truth labels. This formulation jointly determines *which* parameters to update (through $\boldsymbol{\eta}$) and *how* to update them (through $\boldsymbol{\phi}$), while maintaining a limited number of updates for efficiency. Recent studies have scaled and validated this sparse optimization framework in the context of dense LLMs (Ansell et al., 2024), paving the way for broader exploration in sparsity-aware scenarios.

## 3 SPARSITY EVOLUTION FINE-TUNING FOR SPARSE LLMS

Building upon the paradigm of sparse fine-tuning, we propose Sparsity Evolution Fine-Tuning (SEFT), a novel approach that enhances the adaptation of sparse LLMs by allowing the dynamic

Figure 2: An illustration of SEFT for sparse LLMs. The SEFT process consists of: (1) starting with a sparse LLM, along with learnable indices (arrows) and corresponding deltas (green squares) applied to the LLM parameters; (2) dropping obsolete indices (dashed arrows); (3) growing new indices (green arrows); (4) adapting parameters to the target sparsity level (red squares). These steps are repeated every $k$ steps. The final output is a sparse LLM after fine-tuning.

evolution of their sparse topologies during task-specific fine-tuning. Given pre-trained model weights $\boldsymbol{\theta} \in \mathbb{R}^{d_\theta}$, after post-training pruning we obtain a sparse model $\boldsymbol{\theta}' = \boldsymbol{\theta} \odot \boldsymbol{M}_0$, where $\boldsymbol{M}_0 \in \{0,1\}^{d_\theta}$ is a binary mask. For each $i \in \{1, \ldots, d_\theta\}$, $\boldsymbol{M}_0^i = 1$ indicates that parameter $i$ is active (retained) and $\boldsymbol{M}_0^i = 0$ indicates it is inactive (pruned). The sparsity level $\rho$ is defined as the proportion of pruned (inactive) parameters relative to the total number of parameters in the model. We reformulate sparse fine-tuning for such models as:

$$f'(\cdot; \boldsymbol{\theta}') = f(\cdot; \boldsymbol{\theta}' + \boldsymbol{\delta}), \tag{4}$$

where $\boldsymbol{\delta}$ represents the learnable delta that introduces task-specific updates to the sparse model. Note that, SEFT explicitly allows $\boldsymbol{\delta}$ to modify both active ($\boldsymbol{M}_0^i = 1$) and previously pruned ($\boldsymbol{M}_0^i = 0$) weights, enabling the model to recover useful parameters and adapt its sparse structure during fine-tuning. To support this, SEFT introduces two complementary components: *Sparse Topology Evolution*, which explores and activates task-relevant weights, and *Sparsity Adaptation*, which prunes less important weights to maintain the target sparsity level.

This allows efficient and direct acquisition of a performant sparse model through fine-tuning, outperforming the pipeline of pruning a dense model after fine-tuning (see Appendix D).

## 3.1 SPARSE TOPOLOGY EVOLUTION

SEFT periodically updates the indices $\boldsymbol{\eta}$ of non-zero elements in $\boldsymbol{\delta}$ through a *drop-and-grow* mechanism to dynamically evolve the sparse topology during sparse LLMs fine-tuning. Building on empirical evidence from Dynamic Sparse Training (DST) (Mocanu et al., 2018; Evci et al., 2020), which efficiently evolves sparse connectivity during training and consistently outperforms static sparse training, we posit that SEFT's sparse topology evolution enables the recovery and activation of task-relevant connections, thereby improving performance.

**Drop.** During each topology evolution cycle, the smallest updates are dropped from the active set (as in Figure 2 (2)):

$$\boldsymbol{\eta}_{\text{drop}} = \text{argtopk}_{\tau(t)} \left( -|\phi_i| \right), \tag{5}$$

where $|\phi_i|$ denotes the absolute value of the $i$-th element in $\boldsymbol{\delta}$, and $\tau(t)$ determines the number of indices to drop at step $t$. The intuition is that smaller updates reflect minimal task-driven adjustments to the pre-trained model, indicating less contribution and can be safely pruned.

**Grow.** Simultaneously, $\tau(t)$ new weights are added to the active set by selecting those with the largest gradient magnitudes (as in Figure 2 (3)):

$$\boldsymbol{\eta}_{\text{grow}} = \text{argtopk}_{\tau(t)} \left| \nabla_{\boldsymbol{\theta}} \mathcal{L}(\boldsymbol{x}; \boldsymbol{\theta}'_t + \boldsymbol{\delta}) \right|, \tag{6}$$

where $\mathcal{L}$ is the task-specific loss evaluated on the current training batch $\boldsymbol{x}$. Notably, our implementation computes and applies gradients for growth sequentially on a layer-by-layer basis, releasing them immediately afterward. This strategy ensures memory efficiency and avoids the large memory spikes that typically occur in sparse training.

This approach enables the delta vector $\boldsymbol{\delta}$ to dynamically explore beyond the fixed *mask constraints*, i.e., activating updates for both active weights ($\boldsymbol{M}_0^i = 1$) and previously pruned weights ($\boldsymbol{M}_0^i = 0$).

Such that, SEFT improves the model's adaptability, allowing it to recover task-relevant connections and better align its parameter allocation with downstream objectives.

## 3.2 SPARSITY ADAPTATION

While the sparse topology evolution process enables the model to dynamically adapt to downstream tasks, it can lead to a denser model by activating previously inactive weights ($M_0^i = 0$), as shown in Figure 2 (3). To maintain the target sparsity budget, we introduce a sparsity adaptation step that prunes low-importance weights and restores the model to the desired sparsity level, ensuring compliance with the predefined sparsity constraints.

Specifically, low-importance weights are considered redundant and are pruned with minimal impact on overall performance. In this work, we evaluate two importance scoring rules: a magnitude-based score, which is a commonly used pruning metric, and a sensitivity-based score, defined as the product of weight magnitude and its gradient

---

**Algorithm 1** Sparsity Evolution Fine-Tuning (SEFT)

**Input:** Pruned model $\boldsymbol{\theta}'$ with initial mask $\boldsymbol{M_0}$, target sparsity level $\rho$, delta vector $\boldsymbol{\delta}$ initialized as zeros, number of training steps $T$, update frequency $k$.
**Output:** Sparse fine-tuned model $\boldsymbol{\theta}'_T + \boldsymbol{\delta}$.
**for** $t = 1$ to $T$ **do**
    Update $\boldsymbol{\delta}$ using task-specific loss $\mathcal{L}$.
    **if** $t \bmod k = 0$ **then**
        **Sparse Topology Evolution**
        Drop indices $\boldsymbol{\eta}_{\text{drop}}$ based on Eq. 5.
        Grow indices $\boldsymbol{\eta}_{\text{grow}}$ based on Eq. 6.
        Update $\boldsymbol{\delta}$ with $\boldsymbol{\eta}_{\text{drop}}$ and $\boldsymbol{\eta}_{\text{grow}}$.
        **Sparsity Adaptation**
        Calculate important score for each weight.
        Get new mask $M_t$ followed Eq. 7.
        Update sparse weights: $\boldsymbol{\theta}'_t = \boldsymbol{\theta}_t \odot M_t$.
    **end if**
**end for**
**Return:** Sparse fine-tuned model $\boldsymbol{\theta}'_T + \boldsymbol{\delta}$.

---

(Lee et al., 2019; Nowak et al., 2023). As shown in Appendix E.2 and Figure 5 (a), our comparative analysis demonstrates that the sensitivity-based score consistently outperforms the magnitude-based score during the sparsity adaptation phase. Accordingly, we adopt the sensitivity-based score as the default importance metric throughout this work, unless otherwise specified.

**Pruning Decision:** Weights with the smallest importance score $s_i$ are pruned to meet the target sparsity level $\rho$. The new mask $M_t$ is then updated accordingly (as in Figure 2 (4)):

$$M_t^i = \begin{cases} 0, & \text{if } s_i \text{ is among the smallest } \rho \text{ fraction of all } s, \\ 1, & \text{otherwise.} \end{cases} \tag{7}$$

The resulting sparse model is then represented as $\boldsymbol{\theta}'_t = \boldsymbol{\theta}_t \odot M_t$.

This two-stage mechanism, sparse topology evolution followed by sparsity adaptation, enables SEFT to maintain a task-adaptive sparse structure while strictly adhering to the sparsity budget. In practice, SEFT employs accumulated gradient estimates instead of instantaneous gradients, which aims to improve training stability during fine-tuning. An overview of SEFT is provided in Algorithm 1.

## 4 EXPERIMENTS AND RESULTS

**Sparse LLMs.** We evaluate on several widely-used open-source LLMs, including the LLaMA family, DeepSeek, and Mistral. These models are pruned using state-of-the-art post-training pruning methods, such as SparseGPT and Wanda, to achieve the desired sparsity levels. Our focus is primarily on unstructured and N:M sparsity as it allows for fine-grained control over the parameters and has shown strong empirical results in prior works (Frantar & Alistarh, 2023; Sun et al., 2023; Yin et al., 2023). We target highly sparse models (e.g., sparsity $\geq 0.6$), where performance degradation becomes more pronounced, necessitating fine-tuning. Moreover, high sparsity levels are generally more hardware-friendly for practical speedup (Gale et al., 2020), as demonstrated in Section 4.1.3.

**Fine-tuning Tasks.** To comprehensively evaluate the effectiveness of SEFT, we conduct experiments across two categories of fine-tuning tasks:

(1) Performance recovery: This setting assesses the ability of fine-tuning methods to recover performance lost due to pruning. Specifically, models are fine-tuned on a general pretraining dataset, using 30k randomly selected samples from the C4 corpus (Raffel et al., 2020), and evaluated using Wikitext perplexity (PPL) (Merity et al., 2016) and Commonsense Reasoning benchmark (Gao et al., 2021).

(2) Supervised fine-tuning: This setting evaluates the models' ability to learn task-specific information through supervised data: (i) Commonsense Reasoning: Models are fine-tuned on a 170k-sample commonsense reasoning dataset and evaluated on zero-shot performance across seven commonsense reasoning tasks: BoolQ, RTE, HellaSwag, WinoGrande, ARC-e, ARC-c, and OBQA, following the evaluation protocol in (Hu et al., 2023). (ii) MMLU: The Massive Multitask Language Understanding benchmark (Hendrycks et al., 2020) includes 57 tasks spanning diverse domains, such as elementary mathematics, US history, computer science, and law. We adopt a zero-shot evaluation protocol, where models are fine-tuned on an auxiliary training dataset and evaluated on MMLU. (iii) GSM8K (Cobbe et al., 2021): This benchmark consists of grade-school-level math word problems. Models are fine-tuned on the training split of GSM8K and evaluated in a zero-shot setting.

**Baselines.** To evaluate SEFT, we fine-tune the sparse LLMs and compare them with the following sparsity-preserving baselines:

(1) LoRA*: After LoRA fine-tuning, we apply the corresponding post-hoc pruning to restore the desired sparsity level and retain computational efficiency.

(2) SPP: This method fine-tunes sparse models using designed sparsity-preserving adapter, allowing the final model to maintain the target sparsity level after merging (Lu et al., 2024c).

(3) SQFT: We use a sparsity-preserving version from (Munoz et al., 2024) that excludes quantization of the base model. It maintains the target sparsity during fine-tuning by applying a binary mask.

This enables efficient and direct training of performant sparse models, outperforming the prune-after-dense fine-tuning pipeline (see Appendix D). For fair comparison, all fine-tuned LLMs are restored to the target sparsity, and all methods use the same number of trainable parameters, matching to that of LoRA under the corresponding rank configuration. We adopt a fixed rank of 32 as the default setting. Additional experimental details are provided in Appendix J.

### 4.1 OVERALL PERFORMANCE

In this section, we present a comprehensive evaluation of SEFT across a range of fine-tuning scenarios. We begin by analyzing its ability to recover performance after pruning, followed by results on supervised fine-tuning tasks. Finally, we assess its efficiency in terms of memory cost, training speed, and inference latency.

### 4.1.1 PERFORMANCE RECOVERY

We validate the effectiveness of SEFT for performance recovery by fine-tuning sparse LLMs on a pretraining dataset, using 30k randomly sampled examples from C4 (Raffel et al., 2020). To obtain sparse models, we first apply post-training pruning methods including Wanda and SparseGPT to reach a 70% sparsity level.

We use two evaluation metrics: LM-eval, which reports zero-shot accuracy across seven tasks from the EleutherAI LM Harness (Gao et al., 2021), and PPL, which denotes the perplexity on Wikitext-2 (Merity et al., 2016). As shown in Table 1, fine-tuning significantly improves the performance of sparse LLMs, as indicated by lower perplexity and higher LM-eval scores compared to their unfinetuned counterparts. Notably, SEFT demonstrates strong performance recovery across various settings. It generally achieves competitive or better

Table 1: Performance comparison of different methods with and without fine-tuning applied to sparse LLaMA models at a sparsity level of $\rho = 0.7$.

| LLaMA | Method | PPL ($\downarrow$) | LM-eval ($\uparrow$) |
|---|---|---|---|
| V2-7B | Wanda | 75.19 | 34.83 |
| | Wanda+LoRA* | 11.71 | 44.29 |
| | Wanda+SPP | 11.53 | 44.46 |
| | Wanda+SQFT | 11.94 | 44.73 |
| | Wanda+SEFT | **11.19** | **45.61** |
| | SparseGPT | 27.31 | 41.51 |
| | SparseGPT+LoRA* | 12.86 | 45.42 |
| | SparseGPT+SPP | 11.58 | 47.27 |
| | SparseGPT+SQFT | 11.71 | 47.09 |
| | SparseGPT+SEFT | **11.00** | **47.95** |
| V3-8B | Wanda | 120.20 | 34.92 |
| | Wanda+LoRA* | 18.28 | 43.32 |
| | Wanda+SPP | 16.73 | 43.68 |
| | Wanda+SQFT | 17.16 | 43.56 |
| | Wanda+SEFT | **16.17** | **44.55** |
| | SparseGPT | 43.25 | 41.70 |
| | SparseGPT+LoRA* | 17.81 | 46.11 |
| | SparseGPT+SPP | 15.25 | 48.33 |
| | SparseGPT+SQFT | 16.14 | 47.70 |
| | SparseGPT+SEFT | **15.09** | **48.89** |

performance compared to existing baselines, particularly LoRA* and SQFT, across both evaluation metrics. These gains are observed across different model scales and under both pruning strategies.

Table 2: Comparison of fine-tuning methods on sparse LLaMA models at sparsity level $\rho = 0.6$. Results report average and per-task zero-shot accuracy on seven commonsense reasoning tasks. Higher values indicate better performance.

| LLaMA | Method | WG | RTE | OBQA | HS | BoolQ | ARC-e | ARC-c | Avg |
|-------|--------|-----|-----|------|-----|-------|-------|-------|-----|
| V2-7B | Wanda+LoRA* | 73.48 | 54.51 | 29.00 | 51.49 | 70.98 | 72.81 | 39.07 | 55.91 |
| | Wanda+SPP | 73.79 | 54.15 | 27.40 | 51.97 | 71.80 | 70.54 | 37.63 | 55.32 |
| | Wanda+SQFT | 75.61 | 54.87 | 29.00 | 52.86 | 70.46 | 72.51 | 39.07 | 56.34 |
| | Wanda+SEFT | 75.85 | 56.32 | 29.80 | 52.34 | 67.65 | 72.60 | 41.21 | **56.54** |
| | SparseGPT+LoRA* | 73.16 | 53.79 | 28.60 | 51.13 | 77.73 | 72.34 | 38.48 | 56.46 |
| | SparseGPT+SPP | 72.69 | 58.48 | 28.80 | 52.63 | 70.55 | 69.91 | 39.16 | 56.03 |
| | SparseGPT+SQFT | 75.37 | 53.43 | 30.00 | 52.38 | 74.95 | 72.89 | 41.38 | 57.20 |
| | SparseGPT+SEFT | 75.53 | 55.23 | 30.80 | 52.99 | 77.98 | 72.64 | 41.13 | **58.04** |
| V3-8B | Wanda+LoRA* | 73.87 | 56.68 | 27.20 | 50.69 | 69.91 | 71.08 | 39.51 | 55.56 |
| | Wanda+SPP | 73.32 | 54.51 | 27.40 | 50.37 | 70.46 | 70.75 | 37.97 | 54.97 |
| | Wanda+SQFT | 75.84 | 60.26 | 26.40 | 50.85 | 77.61 | 70.24 | 40.01 | 57.32 |
| | Wanda+SEFT | 76.87 | 57.76 | 28.60 | 52.20 | 80.83 | 71.46 | 41.38 | **58.44** |
| | SparseGPT+LoRA* | 79.48 | 57.04 | 29.40 | 53.78 | 83.30 | 74.20 | 43.34 | 60.07 |
| | SparseGPT+SPP | 76.87 | 57.76 | 31.20 | 53.45 | 80.67 | 74.12 | 43.09 | 59.60 |
| | SparseGPT+SQFT | 79.43 | 63.17 | 31.40 | 53.83 | 82.09 | 73.19 | 41.97 | 60.73 |
| | SparseGPT+SEFT | 78.37 | 62.82 | 31.20 | 53.99 | 83.00 | 73.40 | 43.69 | **60.92** |
| V1-13B | Wanda+LoRA* | 77.11 | 59.93 | 33.60 | 56.37 | 82.38 | 76.56 | 44.88 | 61.54 |
| | Wanda+SPP | 80.03 | 62.09 | 32.60 | 57.08 | 76.73 | 75.88 | 44.71 | 61.30 |
| | Wanda+SQFT | 79.48 | 58.84 | 32.40 | 56.84 | 82.54 | 76.05 | 46.42 | 61.79 |
| | Wanda+SEFT | 79.32 | 59.20 | 33.00 | 56.58 | 83.58 | 77.02 | 47.44 | **62.31** |
| | SparseGPT+LoRA* | 78.37 | 59.93 | 31.20 | 55.23 | 79.60 | 76.01 | 43.26 | 60.51 |
| | SparseGPT+SPP | 80.66 | 56.68 | 35.00 | 57.53 | 79.78 | 75.76 | 45.22 | 61.52 |
| | SparseGPT+SQFT | 79.63 | 57.28 | 34.20 | 57.50 | 83.45 | 76.01 | 44.79 | 61.84 |
| | SparseGPT+SEFT | 80.19 | 57.40 | 33.20 | 56.81 | 83.52 | 77.02 | 45.90 | **62.01** |

### 4.1.2 SUPERVISED FINE-TUNING

In this section, we evaluate the effectiveness of SEFT on a widely used supervised fine-tuning tasks-Commonsense Reasoning. Table 2 presents the zero-shot accuracy on seven tasks from the commonsense reasoning benchmark. To begin, the evaluation is performed on post-training sparse models generated by Wanda and SparseGPT at a sparsity level of 60%, which provides a balanced trade-off between efficiency and decent performance comparison on this benchmark. The sparse models are then fine-tuned using SEFT and other sparsity-preserving baselines. We include results for three model sizes: LLaMA2-7B, LLaMA3-8B, and LLaMA1-13B, reporting both per task and average performance.

The results show that SEFT consistently outperforms existing sparsity-preserved fine-tuning methods such as SPP and SQFT across all evaluated models and tasks. Compared to LoRA*, SEFT improves the average accuracy by approximately 1–2%, demonstrating its ability to better adapt sparse models to downstream reasoning tasks. In particular, SEFT achieves strong results in OBQA and ARC-c over other baselines. Overall, these findings highlight SEFT's effectiveness in enhancing the overall performance across commonsense reasoning tasks and model sizes.

### 4.1.3 MEMORY AND COMPUTATION EFFICIENCY

**Memory Efficiency.** In this section, we evaluate the memory overhead of different sparsity-preserving fine-tuning methods using the LLaMA-7B model. Experiments were conducted with sequence lengths ranging from 512 to 2048 at a rank of 32. Figure 3 presents the memory usage of the different methods across various sequence lengths. We find that as the sequence length increases, all methods exhibit a higher memory consumption, which is expected. In particular, SEFT consistently uses less memory than other methods in all settings, and in some cases, SEFT requires approximately half the memory of SQFT.

Compared with LoRA methods, SEFT exhibits substantially lower memory usage as sequence length increases. As shown in the memory consumption breakdown in Appendix C, the majority of SEFT's memory savings comes from reduced *activation memory* during fine-tuning. While LoRA-style adapters introduce additional activation tensors for their adapter pathways, SEFT performs in-place parameter updates without requiring any auxiliary activations. These results highlight SEFT's favorable memory–efficiency profile, offering a stronger balance between resource usage and performance when fine-tuning sparse LLMs.

Figure 3: (a) GPU memory usage (in GB) for SEFT and baseline methods on a Commonsense Reasoning task using an H100 GPU. All results are reported without activation checkpointing.

**Computation Efficiency.** We analyze the inference speedup of the sparse LLM achieved after SEFT fine-tuning, as presented in Table 6 in Appendix B. To assess practical gains, we report end-to-end decoding latency using the LLaMA-V2-7B model running in the DeepSparse inference engine (DeepSparse, 2021) on an Intel Xeon Platinum 8360Y CPU with 36 cores. The results show that fine-tuned sparse LLMs yield substantial inference speedups compared to their dense counterparts, achieving up to 2.5× improvement at 70% sparsity. Moreover, the speedup becomes even more pronounced at higher sparsity levels, demonstrating the potential of SEFT to enable both performance and practical deployment benefits.

## 4.2 Effect on Other Tasks and Architectures

To further evaluate the generality of SEFT beyond standard benchmarks and the LLaMA-based architectures discussed above, we perform experiments on two challenging benchmarks, MMLU and GSM8K, and extend our evaluation to alternative model architectures, including DeepSeek-7B-chat (Bi et al., 2024) and Mistral-7B-v0.1 (Jiang et al., 2023). The MMLU benchmark spans a wide range of academic and professional subjects, testing both factual knowledge and reasoning ability, while GSM8K focuses on grade-school math word problems.

Table 3: Performance evaluation of DeepSeek-7B-chat and Mistral-7B-v0.1 models on MMLU and GSM8K benchmark at a sparsity level of 60% by Wanda pruning method. Higher values indicate better performance.

| Method | | MMLU | GSM8K |
|---|---|---|---|
| **DeepSeek-7B-chat** | LoRA* | 45.57 | 31.46 |
| | SPP | 45.74 | 31.45 |
| | SQFT | 44.23 | 31.97 |
| | SEFT | **46.14** | **32.52** |
| **Mistral-7B-v0.1** | LoRA* | 53.52 | 37.83 |
| | SPP | 53.78 | 39.73 |
| | SQFT | 53.57 | 40.86 |
| | SEFT | **53.97** | **42.91** |

Specifically, we apply the Wanda pruning method to compress DeepSeek-7B-chat and Mistral-7B-v0.1 to a sparsity level of 60%, which offers a balanced setting for meaningful performance comparisons. The pruned LLMs are subsequently fine-tuned using SEFT, LoRA*, SPP, and SQFT, and their performance is evaluated on both the MMLU and GSM8K benchmarks. The results show that SEFT consistently outperforms other methods across both tasks and architectures. For instance, on MMLU, SEFT yields competitive performance, while on GSM8K, it achieves up to around 2% gain on Mistral-7B. These findings underscore the robustness of SEFT in handling diverse tasks and its ability to generalize across different model backbones beyond the LLaMA family. This highlights its potential as a general-purpose sparse fine-tuning method that is both task- and model-agnostic, making it well-suited for real-world deployment scenarios.

## 4.3 Effect on N:M Sparsity

In addition to demonstrating the effectiveness of SEFT under unstructured pruning in previous sections, we further evaluate its applicability to N:M sparsity patterns, which are increasingly supported by modern hardware (e.g., NVIDIA Ampere and Hopper architectures (NVIDIA, 2021)).

To enforce these sparsity pattern constraints, the sparse topology evolution phase in SEFT is restricted to exploring and activating updates only within the set of currently active weights ($M_0 = 1$), preserving the required pattern. Table 4 reports the performance of SEFT under 2:4 and 4:8 sparsity configurations on both LLaMA1-7B and LLaMA3-8B models after fine-tuning pruned models using the Wanda method. The results show that SEFT maintains or even improves performance under N:M sparsity settings, outperforming other sparsity-preserving baselines. These findings extend the applicability of SEFT beyond unstructured pruning, highlighting its versatility in both algorithmic efficiency and hardware-friendly support.

Table 4: Performance evaluation of LLaMA1-7B and LLaMA3-8B models on LM-eval under different N:M sparsity patterns. Higher values indicate better performance.

| Method | LLaMA1-7B | | LLaMA3-8B | |
|---|---|---|---|---|
| | 2:4 | 4:8 | 2:4 | 4:8 |
| LoRA* | 57.35 | 57.71 | 58.25 | 60.04 |
| SPP | 55.16 | 56.77 | 56.55 | 57.81 |
| SQFT | 56.85 | 57.87 | 56.99 | 58.28 |
| SEFT | **58.17** | **58.65** | **59.63** | **60.35** |

## 4.4 IMPACT OF DIFFERENT SPARSITY LEVELS AND RANKS

Table 5 reports LM-eval performance of LLaMA2-7B with sparsity levels $\rho \in \{0.5, 0.6, 0.7, 0.8\}$. SEFT achieves the best results at these sparsity levels, and its advantage grows as $\rho$ increases. At $\rho = 0.8$, for instance, SEFT outperforms the strongest baseline by 1.36 points. This pattern holds across additional tasks and model scales (see Appendix F.1, Table 9). These results suggest that adapting the sparse topology during fine-tuning is particularly effective in high-sparsity regimes compared with fixing a sparsity pattern obtained via post-hoc pruning after fine-tuning.

Table 5: Performance comparison of LLaMA2-7B on LM-eval under different sparsity levels ($\rho$).

| Method | $\rho$=0.5 | $\rho$=0.6 | $\rho$=0.7 | $\rho$=0.8 |
|---|---|---|---|---|
| LoRA* | 60.19 | 56.98 | 49.07 | 39.02 |
| SPP | 59.92 | 56.56 | 49.63 | 40.40 |
| SQFT | 60.37 | 57.84 | 51.58 | 40.70 |
| SEFT | **60.94** | **58.50** | **52.74** | **42.06** |

In Appendix F.2, Table 10 illustrates the impact of the number of fine-tuning parameters on the performance of SEFT. As expected, as the number of trainable parameters increases, SEFT shows potential to improve performance. However, SEFT seems to exhibit a stronger upward trend compared to LoRA base method. For instance, while baseline method shows only marginal improvements at high ranks, SEFT demonstrates consistent performance gains, reaching a much higher accuracy at Rank=64 on commonsense reasoning benchmark.

We further explore the impact of various SEFT configurations, including the drop/grow ratio relative to $\tau(t)$ as defined in Eq.5 and Eq.6 (see Appendix F.3), as well as the frequency of sparse topology evolution (Appendix F.4). An ablation study analyzing the contribution of each SEFT component, such as mask constraints during topology evolution (Appendix E.1) and the role of sparsity adaptation (Appendix E.2), is also provided. In addition, we include a sensitivity analysis of different learning rates in Appendix F.5.

## 5 CONCLUSION

In this work, we propose Sparsity Evolution Fine-Tuning (SEFT), a novel method that dynamically evolves the sparse topology of large language models (LLMs) to better adapt to downstream tasks. By introducing a drop-and-grow mechanism and enabling the reactivation of previously pruned weights, SEFT supports task-specific adaptation while preserving target sparsity constraints. Extensive experiments across a range of benchmarks demonstrate SEFT's effectiveness in both performance recovery and supervised fine-tuning scenarios. Moreover, SEFT achieves these improvements with significantly lower memory and computational costs. These results underscore its potential as a scalable and efficient solution for fine-tuning sparse LLMs in practical applications.

We provide a discussion of related work in Appendix A and present limitations and future directions in Appendix I.

## 6 ETHICS STATEMENT

This paper contributes to the development of efficient fine-tuning methods for large language models (LLMs), addressing the challenges posed by their substantial computational and performance demands and enhancing their feasibility for real-world deployment. While our contributions do not inherently lead to negative societal impacts, we encourage the community to remain mindful of potential ethical and practical implications when extending or applying our research.

## 7 REPRODUCIBILITY STATEMENT

To ensure reproducibility, we include our source code in the Supplementary Material and will release it publicly at camera-ready. The implementation details of our method are described in Section 3.1 and Algorithm 1. Comprehensive information on training configurations, hyperparameters, and datasets is provided in Appendix J.

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

APPENDIX

## A   RELATED WORK

### A.1   LLM PRUNING

In the era of large language models (LLMs), their enormous sizes pose significant challenges for real-world deployment, including increased computational and memory demands. To address these challenges recent research has increasingly focused on post-training pruning methods, which start from pre-trained networks and remove redundant parameters to reduce model size and complexity (Ma et al., 2023; Naveed et al., 2023; Zhu et al., 2024). For instance, SparseGPT utilizes second-order information to solve a layer-wise reconstruction problem, enabling the effective pruning of large models (Frantar & Alistarh, 2023). Similarly, Wanda introduces a pruning metric that considers both weight magnitude and corresponding activations (Sun et al., 2023). More recently, layer-wise prunability has been introduced as a technique to enhance traditional pruning methods by adaptively allocating sparsity across layers based on their individual importance (Yin et al., 2023; Lu et al., 2024b). However, existing methods often struggle to maintain satisfactory performance, especially under high sparsity levels. Moreover, state-of-the-art LLM pruning techniques heavily rely on calibration data, which may inadvertently remove connections critical for specific downstream tasks, leading to suboptimal performance. To preserve the efficiency benefits of pruning while maintaining accuracy, we propose SEFT, a fine-tuning method that maintains sparsity throughout training and enables the dynamic evolution and recovery of sparse connectivity. Unlike the method (Yi-Lin Sung, 2021; Jonathan Frankle, 2019) that rely on a dense model and update only a fixed subset of parameters during training, SEFT continually repairs task-relevant connections.

## A.2 Parameter-efficient Fine-tuning

Parameter-Efficient Fine-Tuning (PEFT) has gained significant attention for its ability to adapt large language models (LLMs) to downstream tasks while significantly reducing computational and memory costs (Mangrulkar et al., 2022; Ding et al., 2023; Han et al., 2024). A notable PEFT approach, Low-Rank Adaptation (LoRA) (Hu et al., 2021), and its variants introduce trainable layer-wise low-rank decomposition matrices into network, allowing efficient parameter updates with minimal overhead (Xu et al., 2023; Zhao et al., 2024; Dettmers et al., 2024). More recent methods attempt to further improve efficiency by selecting a subset of base-model weights for fine-tuning Song et al. (2024); Yang et al. (2024); however, these approaches still require maintaining a dense base model during training. However, when fine-tuning sparse LLMs, a challenge that arises is that merging dense adapters with sparse weights leads to the overall loss of sparsity, which negates the efficiency benefits of sparse models. Recent studies such as SPP (Lu et al., 2024c) and SQFT (Munoz et al., 2024) have extended LoRA to support sparse LLMs by incorporating masking mechanisms, thereby preserving sparsity during fine-tuning.

In contrast, sparse fine-tuning, which updates only a subset of the model's original parameters with minimal architectural changes, has emerged as a practical strategy for PEFT. However, its practical memory and computational benefits often diminish at large scale. Methods such as SIFT (Song et al., 2024), PaFi (Baohao Liao, 2023), SHiRA (Bhardwaj et al., 2024) and SpIEL (Ansell et al., 2024) impose sparse updates, yet still rely on dense pretrained models and result in dense models after fine-tuning. Other approaches, including SMT (He et al., 2025) and GPS (Zhang et al., 2024), improve efficiency by selecting gradients per neuron. Nevertheless, they depend on gradient-based warm-up phases and additional binary masking. These mechanisms add overhead from mask storage, dense model dependency, or fixing weight selection in advance, making them less suitable for sparse language model fine-tuning across diverse downstream tasks. In contrast, SEFT inherits the advantages of sparsity while dynamically recovering task-relevant connectivity and maintaining an efficient sparse topology throughout training.

## A.3 Dynamic Sparse Training

Dynamic Sparse Training (DST) is a paradigm that maintains a small fraction of active parameters throughout training by starting with a sparse neural network and dynamically evolving its sparse connectivity using a prune-and-regrow strategy (Mocanu et al., 2018; Evci et al., 2020; Liu et al., 2023; Wu et al., 2025). It was first well established by (Mocanu et al., 2018) through the Sparse Evolutionary Training (SET) algorithm, which demonstrated superior performance compared to static sparse neural networks by dynamically evolving sparse connectivity during training. Recent advancements have explored diverse pruning criteria, such as magnitude-based (Evci et al., 2020; Liu et al., 2021), weight-balanced (Mocanu et al., 2018), and gradient-based pruning (Yuan et al., 2021), alongside regrowth strategies guided by randomness (Mostafa & Wang, 2019; Xiao et al., 2022), momentum (Dettmers & Zettlemoyer, 2020), and gradient information (Evci et al., 2020). DST has been effectively applied and widely adopted in various domains, including reinforcement learning (Graesser et al., 2022), features selection (Sokar et al., 2022; Atashgahi et al., 2022), and image segmentation (Wu et al., 2024). Building on this foundation, recent work (Ansell et al., 2024) was the first to scale DST to fine-tuning LLMs. Our work extends this further to sparse LLMs by introducing dynamic topology evolution and adaptation, enabling efficient and task-specific fine-tuning for downstream applications.

# B Inference Speedup on CPU

While support for unstructured sparsity on modern GPUs remains relatively limited, there is growing attention toward enabling such capabilities. For example, Cerebras has developed specialized hardware designed to support unstructured sparsity at scale (Lie, 2023; Thangarasa et al., 2024). Additionally, NVIDIA's Ampere and Hopper architectures have introduced hardware-level support for structured sparsity (e.g., 2:4 patterns), enabling modest acceleration through sparse tensor cores (NVIDIA, 2021). These efforts reflect a broader trend toward making sparsity-aware training and inference feasible on GPU hardware.

Despite these advancements, unstructured sparsity has shown immediate and practical benefits on non-GPU platforms such as CPUs and custom accelerators. For instance, FPGA-based accelerators for sparse RNNs have achieved notable gains in speed and energy efficiency by fully utilizing embedded multipliers. A particularly prominent example is DeepSparse [1], which efficiently deploys large-scale sparse models like BERT on modern Intel CPUs. DeepSparse reports up to 10× model compression with less than 1% accuracy loss, 10× CPU inference speedup with under 2% drop, and as much as 29× speedup with under 7.5% accuracy degradation.

Table 6: End-to-end speedup of LLaMA-V2-7B under different sparsity levels with the DeepSparse inference engine.

| Sparsity | Dense | 40% | 50% | 60% | 70% | 80% |
|---|---|---|---|---|---|---|
| Latency(ms ) | 206.36 | 179.22 | 113.69 | 95.18 | 82.29 | 63.52 |
| Throughput | 4.84 | 5.58 | 8.79 | 10.50 | 12.15 | 15.74 |
| Speedup | 1.0× | 1.2× | 1.8× | 2.2× | 2.5× | 3.3× |

Motivated by these developments, we evaluate actual inference speedup using the DeepSparse engine (DeepSparse, 2021). Specifically, we measure end-to-end decoding latency for the LLaMA-V2-7B model running on an Intel Xeon Platinum 8360Y CPU with 36 cores. Results show that sparse LLMs fine-tuned with SEFT achieve substantial speedups over their dense counterparts, reaching up to 2.5× improvement at 70% sparsity. Furthermore, the benefit becomes even more pronounced at higher sparsity levels, achieving approximately 4× speedup at 90% sparsity, highlighting the promise of extreme sparsity for efficient inference. These findings underscore the importance of maintaining sparsity throughout the fine-tuning process and point toward the potential of leveraging sparsity-aware GPU operations in future deployment scenarios.

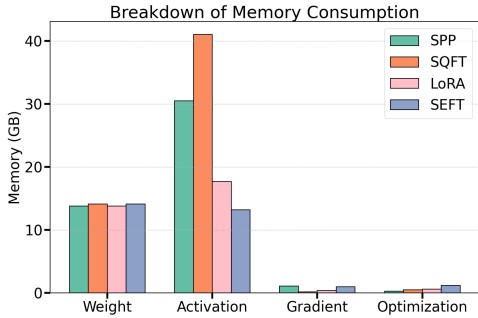 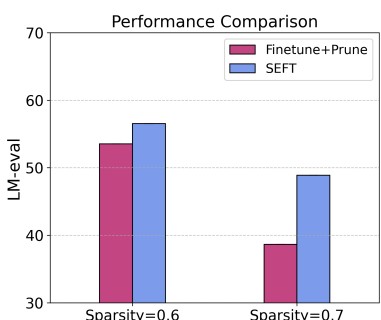

Figure 4: (a) Breakdown of memory consumption for different methods when fine-tuning the LLaMA-2 7B model. (b) LM-eval performance of LLaMA-2 7B under varying sparsity levels, comparing SEFT with baseline approaches.

## C    MEMORY USAGE COMPARISON

We measure end-to-end GPU memory consumption and its component-wise breakdown (parameters, activations, gradients, and optimizer states) on fine-tuning the LLaMA-2 7B model. Unless otherwise noted, we use batch size $= 1$, input sequence length $= 1024$, and *disable* gradient checkpointing. For the compared methods (LoRA, SPP, and SQFT), we apply the same settings. The results are shown in Figure 4 (a).

SEFT exhibits substantially lower *activation* memory than the baselines, whereas its *gradient* and *optimizer* memory are slightly higher. The latter is attributable to temporary buffers used during the sparse topology evolution step. Overall, SEFT achieves the lowest total memory footprint. Moreover, because activation memory scales roughly linearly with sequence length, SEFT's advantage increases at longer contexts; hence we expect even larger savings as the sequence length grows.

---

[1] https://github.com/neuralmagic/deepsparse

## D COMPARISON WITH POST-HOC PRUNING

Figure 4 (b) reports LM-eval performance of fine-tuning LLaMA-2 7B model across sparsity levels, comparing the standard pipeline—LoRA fine-tuning on a *dense* model followed by post-hoc pruning—with *SEFT*, which fine-tunes an already sparse model. SEFT consistently outperforms the *prune after dense fine-tuning* pipeline, with particularly large gains at higher sparsity (e.g., >10% LM-eval points at 70% sparsity).

These results indicate that SEFT is not only more *efficient*, by maintaining a sparse base model throughout fine-tuning, but also more *effective* at aligning the sparse topology with downstream tasks. This provides a clear motivation for adopting pruning *and* sparse fine-tuning as an integrated strategy, rather than fine-tuning densely and pruning afterward.

## E ABLATION STUDY

We conduct a series of ablation studies to better understand the impact of key components in SEFT. Specifically, we analyze (1) the effect of removing the mask constraint during sparse topology evolution, (2) the role of sparsity adaptation in maintaining the target sparsity level, and (3) the benefit of using a sensitivity-based pruning criterion over a magnitude-based one.

Table 7: Performance comparison of LLaMA2-7B and LLaMA3-8B models with and without mask constraints for SEFT during sparse topology evolution at different sparsity levels ($\rho$).

| Method | | LLaMA2-7B | | LLaMA3-8B | |
|---|---|---|---|---|---|
| | | $\rho = 0.6$ | $\rho = 0.7$ | $\rho = 0.6$ | $\rho = 0.7$ |
| **LM-eval** | w. constraint | 55.98 | 48.36 | 58.17 | 47.12 |
| | w/o. constraint | **56.54** | **48.87** | **58.44** | **47.44** |
| **MMLU** | w. constraint | 45.74 | 31.18 | 50.03 | 39.59 |
| | w/o. constraint | **45.95** | **33.51** | **50.33** | **39.87** |

### E.1 EFFECT ON MASK CONSTRAINTS

In Section 3.1, we introduced how SEFT dynamically evolves the sparse topology during fine-tuning through a drop-grow strategy. Specifically, SEFT enables the delta vector $\eta$ to explore and activate updates for both active weights ($M_0 = 1$) and inactive weights ($M_0 = 0$) in sparse LLMs, enabling dynamic sparse topology evolution. To assess the necessity of reactivating inactive weights, we compare SEFT against a constrained version where the delta vector $\eta$ is restricted to updating only the active weights ($M_0 = 1$) in sparse LLMs. We refer to SEFT as *w/o. constraint* and the constrained version as *w. constraint*.

The results in Table 7 show that for both LLaMA2-7B and LLaMA3-8B models pruned to 60% and 70% sparsity levels using Wanda, SEFT with sparse topology evolution (*w/o. constraint*) consistently outperforms the constrained version on both the commonsense reasoning and MMLU benchmarks. These findings demonstrate that removing the mask constraint, which allows updates to inactive weights, facilitates sparse topology evolution in sparse LLMs. This evolution aligns the model more effectively with task-specific demands, resulting in improved overall performance and mitigating the degradation introduced by pruning methods.

### E.2 EFFECT ON SPARSITY ADAPTION

In Section 3.2, we proposed a sparsity adaptation process to restore the model to the desired sparsity level, as sparse topology evolution in SEFT without a mask constraint may result in a denser model by reactivating previously inactive weights. We conducted experiments to evaluate the effectiveness of sparsity adaptation on LLaMA1-7B, LLaMA2-7B, and LLaMA3-8B models pruned to 60% sparsity using Wanda, comparing SEFT with and without sparsity adaptation. As shown in Table 8, the results indicate that without sparsity adaptation, the sparse LLMs become slightly denser after fine-tuning,

Table 8: Performance comparison of LLaMA1-7B, LLaMA2-7B, and LLaMA3-8B models with and without sparsity adaptation for SEFT during sparse topology evolution at a sparsity level of 60%.

| Method | | LM-eval | Final sparsity |
|---|---|---|---|
| **LLaMA1-7B** | w/o. adapt | 58.85 | 0.596 |
| | w. adapt | 58.84 | 0.600 |
| **LLaMA2-7B** | w/o. adapt | 56.45 | 0.596 |
| | w. adapt | 56.54 | 0.600 |
| **LLaMA3-8B** | w/o. adapt | 58.60 | 0.597 |
| | w. adapt | 58.44 | 0.600 |

with a sparsity level of approximately 59.6%. In contrast, with sparsity adaptation, the models maintain their original sparsity levels while achieving comparable performance.

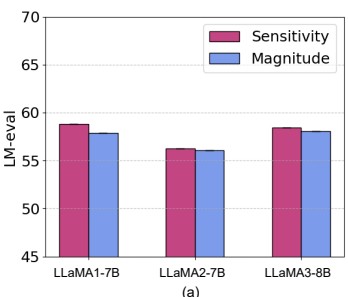 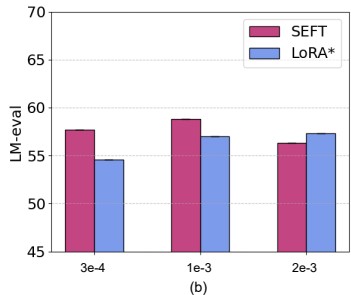

Figure 5: (a) LM-eval comparison of LLaMA1-7B, LLaMA2-7B, and LLaMA3-8B models with sensitivity-based and magnitude-based sparsity adaptation in SEFT fine-tuning at 60% sparsity using Wanda pruning. (b) LM-eval comparison of LLaMA1-7B models at 60% sparsity using Wanda pruning for LoRA* and SEFT fine-tuning across different learning rates.

**Importance metrics.** In our work, we compare two scoring metrics: (i) *magnitude-based*, $s_i = |\theta_i|$, which is task-agnostic and inexpensive to compute; and (ii) *sensitivity-based*, $s_i = |\theta_i \nabla_{\theta_i} \mathcal{L}|$, which accounts for both parameter size and its contribution to the loss (Lee et al., 2019; Wu et al., 2023; Nowak et al., 2023).

In Figure 5 (a), we evaluate the effectiveness of sensitivity-based criterion by comparing it to the commonly used magnitude-based criterion for pruning back to the target sparsity. Magnitude-based criterion are widely used in dynamic sparse training (DST) scenarios. Our results demonstrate that sensitivity-based criterion consistently outperform magnitude-based criterion. This advantage is likely due to the sensitivity-based approach accounting not only for the magnitude of the weights but also for gradient information, which reflects their future importance during training.

# F    SENSITIVITY ANALYSIS

We conduct a detailed sensitivity analysis to examine how SEFT responds to different hyperparameter choices. Specifically, we study (1) the impact of sparsity levels on fine-tuning performance, (2) the effect of the number of trainable parameters, (3) the influence of drop rate and (4) evolution frequency in sparse topology evolution, and (5) the sensitivity to learning rate configurations. These analyses offer insights into SEFT's robustness and adaptability across settings, and help guide optimal hyperparameter choices for various deployment scenarios.

### F.1 Impact on Sparsity Level

Table 9 presents the performance evaluation of LLaMA2-7B and LLaMA3-8B models using Wanda pruning under varying sparsity levels. The results show that as the sparsity level increases, SEFT consistently outperforms LoRA* by larger margins. For instance, at a sparsity level of 70%, SEFT achieves significant improvements in both LM-eval and MMLU scores compared to LoRA*. Notably, on LLaMA3-8B, the relative gain in MMLU reaches approximately 8.69 points (39.87 vs. 31.18). This trend highlights that SEFT's ability to dynamically adapt the sparse topology during fine-tuning is particularly effective in high-sparsity scenarios. The primary reason for this is that post-training pruning methods like Wanda tend to degrade more significantly at higher sparsity levels, resulting in greater performance gaps. By leveraging sensitivity-based criterion, SEFT is able to better identify and optimize critical parameters, maintaining strong performance even under extreme sparsity conditions where LoRA* struggles to achieve comparable results.

Table 9: Performance comparison of LLaMA2-7B and LLaMA3-8B models on LM-eval and MMLU under different sparsity levels ($\rho$).

| Method | | LLaMA2-7B | | | LLaMA3-8B | | |
|---|---|---|---|---|---|---|---|
| | | $\rho$=0.5 | $\rho$=0.6 | $\rho$=0.7 | $\rho$=0.5 | $\rho$=0.6 | $\rho$=0.7 |
| **LM-eval** | LoRA* | **60.12** | 56.19 | 46.36 | 62.90 | 55.56 | 44.82 |
| | SEFT | 59.30 | **56.54** | **48.87** | **63.04** | **58.44** | **47.44** |
| **MMLU** | LoRA* | 48.21 | 42.29 | 25.37 | 57.40 | 49.18 | 31.18 |
| | SEFT | **48.90** | **45.95** | **33.51** | **57.51** | **50.33** | **39.87** |

Table 10: Performance comparison of LLaMA1-7B fine-tuned on the commonsense reasoning dataset at a sparsity level of 60%. The results compare different numbers of fine-tuning parameters and report the average zero-shot accuracy across seven tasks from the commonsense reasoning benchmark.

| Method | Rank=8 | Rank=16 | Rank=32 | Rank=64 |
|---|---|---|---|---|
| LoRA* | 56.81 | 57.13 | 57.32 | 57.17 |
| SEFT | 57.74 | 57.61 | 58.84 | 59.69 |

### F.2 Impact on Number of Fine-tuning Parameters

In this section, we analyze how the number of fine-tuning parameters impacts the performance of SEFT and LoRA* methods. The comparison is based on different numbers of fine-tuning parameters, determined by the ranks in LoRA. For SEFT, the number of fine-tuning parameters is aligned with the corresponding parameter count of LoRA ranks to ensure a fair comparison.

As shown in Table 10, the performance of both LoRA* and SEFT improves as the number of fine-tuning parameters increases, which is expected. However, SEFT exhibits a significantly stronger upward trend compared to LoRA*. For instance, while LoRA* shows only marginal improvements at higher ranks (e.g., from 57.32 at Rank=32 to 57.17 at Rank=64), SEFT demonstrates consistent performance gains, reaching a much higher accuracy of 59.69 at Rank=64.

This trend indicates that SEFT is better equipped to utilize the additional fine-tuning capacity to adapt sparse LLMs to downstream tasks. By dynamically evolving the sparse topology during fine-tuning, SEFT effectively redistributes its parameter budget toward critical updates, enabling it to achieve more substantial improvements as the number of parameters increases. Overall, SEFT consistently outperforms LoRA* across all ranks, with the performance gap widening at higher parameter counts. These results highlight SEFT's scalability and its ability to efficiently leverage additional fine-tuning parameters for enhanced task-specific performance.

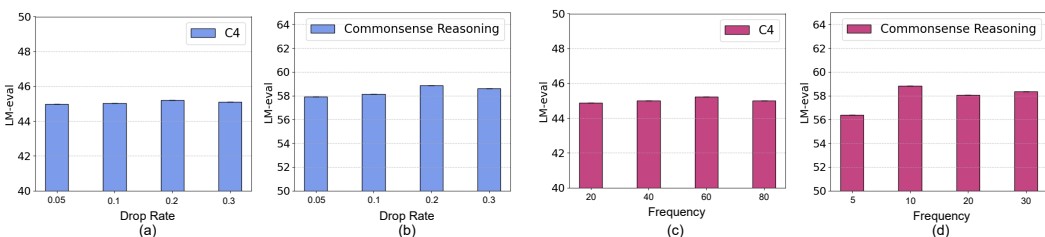

Figure 6: Impact of drop rate during sparse topology evolution on LM-eval evaluation of LLaMA1-7B fine-tuning for (a) C4 and (b) commonsense reasoning datasets. Furthermore, the impact of update frequency during sparse topology evolution is evaluated on LM-eval for LLaMA1-7B fine-tuning on (c) C4 and (d) commonsense reasoning datasets.

### F.3 IMPACT ON DROP RATE

To investigate the effect of varying drop rates, which determine the fraction of weights pruned during each update, we conducted experiments using different initial drop rates for SEFT fine-tuning on the C4 and Commonsense Reasoning datasets. Drop rates of 0.05, 0.1, 0.2, 0.3 were evaluated, along with a cosine schedule decay for the drop/grow ratio, as proposed in (Evci et al., 2020).

As shown in Figure 6 (a) (b), the results demonstrate that a drop rate of 0.2 consistently achieves the best performance across these tasks. Lower drop rates, such as 0.05, fail to sufficiently adapt the sparse topology, limiting performance improvements, while higher drop rates, such as 0.3, can overly perturb the model, disrupting training convergence. The balanced drop rate of 0.2 was adopted as the default setting for all subsequent experiments, as it effectively balances adaptation and stability during sparse topology evolution.

### F.4 IMPACT ON UPDATE FREQUENCY $k$

The frequency of sparse topology evolution, which determines how often the sparse topology is updated during training, is another critical factor. We evaluated different drop/grow frequencies for fine-tuning on the C4 dataset (10, 20, 40, 80 steps) and the Commonsense Reasoning dataset (5, 10, 20, 30 steps).

As shown in Figure 6 (c) (d), the optimal frequency varies by task, reflecting differences in training dynamics and the rate at which sparse topology adaptation benefits performance. For example, on the C4 dataset, the best performance was achieved with an update frequency of 60 steps, while for the Commonsense Reasoning dataset, the optimal frequency was 10 steps. These frequencies were adopted as the default settings in our main experiments.

The results also reveal that overly frequent updates can introduce excessive instability, preventing the model from converging effectively. Conversely, infrequent updates reduce the model's ability to adapt its sparse topology, thereby limiting the benefits of sparsity. Striking the right balance in the frequency of topology evolution is crucial for maximizing SEFT's performance across diverse tasks.

### F.5 IMPACT ON LEARNING RATE

In this section, we analyze how the learning rate affects the performance of the sparse fine-tuning method SEFT and the LoRA-based fine-tuning method LoRA*. As shown in Figure 5 (b), we conduct experiments on LLaMA1-7B under a sparsity level of 0.6, using the commonsense reasoning dataset. The results report the zero-shot accuracy across seven tasks from the commonsense reasoning benchmark. From the figure, we observe that LoRA* generally performs better with a larger learning rate, achieving the best performance at a learning rate of 2e-3. In contrast, SEFT performs best at a slightly smaller learning rate of 1e-3. Interestingly, under smaller learning rates, SEFT achieves more significant improvements compared to LoRA*.

This phenomenon may be attributed to the fundamental differences in how the two methods update sparse LLMs. SEFT directly updates the original weights of the sparse LLMs, enabling more precise

Table 11: Performance comparison of fine-tuning methods on 70% sparse LLaMA models. We report average accuracy across 7 zero-shot tasks (LM-eval ↑) and perplexity (PPL ↓; lower is better).

| Model | Method | PPL (↓) | LM-eval (↑) |
|-------|--------|---------|-------------|
| LLaMA V2-7B | Wanda+LoRA | 10.82 | 45.08 |
| | Wanda+SEFT | 11.19 | 45.61 |
| | SparseGPT+LoRA | 10.53 | 47.80 |
| | SparseGPT+SEFT | 11.00 | 47.95 |
| LLaMA V3-8B | Wanda+LoRA | 16.12 | 44.76 |
| | Wanda+SEFT | 16.17 | 44.55 |
| | SparseGPT+LoRA | 17.88 | 47.19 |
| | SparseGPT+SEFT | 15.09 | 48.89 |

control and better adaptation, especially under smaller updates at lower learning rates. This behavior is similar to dense full fine-tuning, where the original model weights are explicitly updated to reflect task-specific requirements. In contrast, LoRA-based fine-tuning approximates updates through low-rank matrices, indirectly influencing the original weights of the sparse LLMs. This indirect update mechanism allows for more flexible and larger updates, as the original weights remain intact and act as a safeguard against errors introduced by the approximation.

These results highlight the critical role of learning rate selection in optimizing the performance of sparse fine-tuning methods. To ensure a fair comparison, we performed a grid search for both fine-tuning approaches.

## G    COMPARISON WITH ORIGINAL LoRA

In the main paper, to ensure a fair comparison, all fine-tuned LLMs were restored to the same target sparsity level after fine-tuning. In this section, we present the original results for LoRA, which do not include sparsity restoration. In other words, the fine-tuned LLMs remain dense after merging the low-rank matrices with the original model weights.

Tables 11 and 12 provide a detailed comparison between SEFT and LoRA across three datasets: C4, Commonsense Reasoning. The results demonstrate that SEFT achieves comparable performance to LoRA, even when the latter uses dense connections in the LLMs. However, SEFT maintains the sparse topology at the original sparsity level, preserving the efficiency of the sparse model.

These findings underscore the effectiveness and efficiency of SEFT, as it not only matches the performance of LoRA fine-tuning with dense models but also retains the computational advantages of sparsity, making it a more practical solution for sparse fine-tuning of large language models, , as discussed in detail in Section B.

Table 12: Performance comparison of different fine-tuning methods applied to various sparse LLaMA models at a sparsity level of ($\rho = 0.6$). Results include the average accuracy of zero-shot evaluation across seven tasks from the commonsense reasoning benchmark.

| LLaMA | Method | LM-eval |
|-------|--------|---------|
| V2-7B | Wanda+LoRA | 56.66 |
| | Wanda+SEFT | 56.54 |
| | SparseGPT+LoRA | 58.24 |
| | SparseGPT+SEFT | 58.04 |
| V3-8B | Wanda+LoRA | 59.51 |
| | Wanda+SEFT | 58.44 |
| | SparseGPT+LoRA | 60.66 |
| | SparseGPT+SEFT | 60.92 |
| V1-13B | Wanda+LoRA | 61.09 |
| | Wanda+SEFT | 62.31 |
| | SparseGPT+LoRA | 61.99 |
| | SparseGPT+SEFT | 62.01 |

## H    DISCUSSION

Preserving sparsity during and after the fine-tuning of sparse LLMs is essential for maintaining computational efficiency, particularly in resource-constrained scenarios. Methods like LoRA, originally

designed for resource-efficient fine-tuning of dense pre-trained models, fail to preserve sparsity after merging, limiting their effectiveness for sparse LLMs fine-tuning. Recent approaches, such as SPP (Lu et al., 2024c), extend LoRA to support sparse LLMs by incorporating masking mechanisms to preserve sparsity. However, SPP enforces a fixed sparse topology across all tasks, which limits its adaptability to the specific requirements of downstream applications.

Sparse fine-tuning was first scaled to dense LLM fine-tuning in SpIEL (Ansell et al., 2024). Unlike LoRA-based methods, which use trainable low-rank matrices to parameterize adaptations, fine-tuning directly updates a small fraction of model weights through updates and their corresponding index vectors. In this paper, we extend this concept to sparse LLMs fine-tuning by enabling the updates to dynamically evolve and adapt in sparse LLMs. While SEFT builds upon the core concept of sparse fine-tuning, it introduces several key innovations tailored to sparse LLMs:

(1) Sparse setting. Unlike SpIEL and SMT (He et al., 2025), which operates on dense models, SEFT is designed for fine-tuning already-pruned sparse LLMs. This setting imposes strict sparsity constraints that require careful management during training.

(2) Update flexibility sparsity adaptation. SEFT allows updates to zero-valued (previously pruned) weights, enabling recovery of important connections for downstream tasks. It also incorporates a sparsity adaptation step to ensure the model maintains its target sparsity after each update cycle—both of which are absent in SpIEL and SMT.

(3) The main motivation of SEFT is to recover pruned connections that are relevant to specific downstream tasks. As many state-of-the-art LLM pruning methods rely heavily on calibration data and may remove connections crucial for task-specific generalization. While the pruning-and-growth mechanism in SpIEL and the parameter selection in SMT are designed to choose which parameters to update, the pruning-and-growth mechanism in SEFT is primarily used to dynamically recovery the pruned but task-relevant weights during finetuning. Additionally, SMT selects the most influential sparse sub-matrices only during the warm-up phase, whereas SEFT applies its pruning-and-growth mechanism to select and recover task-relevant parameters throughout training.

These distinctions make SEFT a more tailored solution for fine-tuning sparse LLMs across varying sparsity levels.

## I  LIMITATIONS AND FUTURE WORK

While our work highlights the effectiveness of SEFT in fine-tuning sparse LLMs and enhancing their performance on downstream tasks, there are certain limitations that warrant further investigation.

One key limitation of SEFT lies in the need to compute full dense gradients during the sparse topology evolution phase, from which sparse delta updates are subsequently extracted. Although our implementation mitigates memory pressure by computing and applying gradients sequentially on a layer-by-layer basis and releasing them immediately afterward—thereby ensuring memory efficiency and avoiding large spikes—this design still limits computational efficiency. In particular, it does not fully leverage the potential speed and resource benefits of sparsity on GPUs, since the dense gradient matrix must first be calculated before isolating the sparse updates.

Writing an efficient CUDA kernel tailored for dynamic sparse training operations is an area of ongoing work. Such a kernel would enable direct computation of sparse updates without requiring operations on the full gradient matrix, significantly reducing memory usage and computational overhead. Once optimized, this approach could fully unlock the potential of SEFT and dynamic sparse training paradigms, making them more practical and efficient for large-scale federated and distributed learning scenarios.

## J  EXPERIMENTAL SETTINGS

For training 7B models, we use a learning rate of 1e-3, except for Mistral-7B, where we use 5e-5. For 8B models, we adopt a learning rate of 3e-4. Gradient accumulation steps are set to 128, and we apply cosine learning rate decay throughout training. We use the AdamW optimizer with the default

settings provided by the Transformers library, and no weight decay is applied. For baseline methods, we primarily build upon the official implementations of SPP[2] and SQFT[3].

All models are fine-tuned using post-training pruned LLMs obtained via SparseGPT and Wanda under various sparsity patterns and levels. We use fixed hyperparameters across experiments and do not perform tuning for specific sparsity configurations. We follow the dataset configurations in (Li et al., 2025), with details summarized in Table 13.

Table 13: Hyperparamters used of SEFT for fine-tuning on various benchmarks.

| Benchmarks | Commonsense Reasoning | MMLU | GSM8K |
|---|---|---|---|
| Train Samples | 170K | 99.8K | 7.4K |
| Test Samples | 22.4K | 14K | 1.3K |
| Max Length | 512 | 512 | 512 |
| Training Epoch | 1 | 1 | 5 |
| Drop Rate | 0.2 | 0.2 | 0.2 |
| Frequency | 10 | 60 | 10 |

---

[2]https://github.com/Lucky-Lance/SPP
[3]https://github.com/IntelLabs/Hardware-Aware-Automated-Machine-Learning

