# OpenReview forum: "Leave it to the Specialist: Repair Sparse LLMs with Sparse Fine-Tuning via Sparsity Evolution"
_ICLR.cc/2026/Conference — Submitted to ICLR 2026_

### Official Review · Reviewer_dRk7 · 2025-10-27

**Soundness:** 3
**Presentation:** 3
**Contribution:** 3
**Rating:** 6
**Confidence:** 2

**Summary:**

The paper proposes Sparsity Evolution Fine-Tuning (SEFT), a new method tailored for already-pruned large language models. It dynamically evolves the sparse topology during finetuning via a weight drop-and-grow strategy coupled with a sensitivity-driven pruning criterion, so task-relevant connections are restored while the overall sparsity budget is strictly preserved. Extensive experiments on multiple LLaMA, DeepSeek, and Mistral models across diverse benchmarks demonstrate that SEFT consistently outperforms prior baselines in both accuracy and memory-computation efficiency.

**Strengths:**

1. The proposed drop–grow–adapt procedure intuitively tackles the challenges of sparse fine-tuning: unlike LoRA, it allows high-rank weight updates while strictly preserving high sparsity, and its reduced memory footprint further boosts its potential for widespread adoption.
2. The manuscript is clearly written and easy to follow.

**Weaknesses:**

W1. Continuously updating the sparsity masks adds extra computation—especially because full dense gradients are computed during the sparse-topology-evolution phase—yet the paper does not report the actual training-time overhead versus LoRA, making it difficult to gauge the true computational cost.

W2. Although sparsification is promising, in practice weight quantization is often more useful, and state-of-the-art methods can reach 1–2-bit precision with minimal accuracy loss (roughly comparable to a 90 % sparsity level). Taking this into account, investigating whether the method can be combined with quantization would be an important consideration for practical applicability.

**Questions:**

Please clarify the concerns raised in W1 and W2.

Minor
• Typo at L885: “The results, presented in Table 7, show that …”.

---

> ### Author Response · Authors · 2025-11-21
>
> Thank you so much for your time and effort in reviewing our paper. We provide point-wise responses to address your comments below.
>
> > W1. Continuously updating the sparsity masks adds extra computation—especially because full dense gradients are computed during the sparse-topology-evolution phase—yet the paper does not report the actual training-time overhead versus LoRA, making it difficult to gauge the true computational cost.
>
> A: Thank you for pointing out this important concern. We agree that updating the sparse topology during training does introduce some computational overhead. However, in our implementation, **instead of updating the topology at every step, we adopt an efficient strategy in which SEFT performs topology updates only at regular intervals.** This keeps the additional computation minimal throughout the entire training process, as reflected in the average per-step runtime comparisons reported in Table 10. For example, on the MMLU downstream task, the topology is updated every 60 steps. As shown in Section F.4 and Figure 6, **using a reasonable update frequency does not significantly affect performance.**
>
> Importantly, even after accounting for this overhead, SEFT remains consistently faster than all baselines throughout the full training process. Thus, including the cost of topology evolution does not diminish SEFT’s overall efficiency advantage over other methods.
>
> Table 11. Comparison of average per-step running time (in seconds) (**↓**) for LLaMA2-7B and LLaMA3-8B.
> |model|LLaMA2-7B| LLaMA3-8B
> |-|-|-|
> |LoRA| 20.7| 21.6|
> |SPP| 55.6| 58.9|
> |SQFT| 38.5| 40.2|
> |SEFT wo/ Evolution| 19.2| 19.8|
> |SEFT w/ Evolution| 19.6| 20.2|
>
>
> > W2. Although sparsification is promising, in practice weight quantization is often more useful, and state-of-the-art methods can reach 1–2-bit precision with minimal accuracy loss (roughly comparable to a 90 % sparsity level). Taking this into account, investigating whether the method can be combined with quantization would be an important consideration for practical applicability.
>
> A: We appreciate this insightful comment. We agree that, alongside sparsification, weight quantization is another important and widely used compression technique, and understanding how SEFT interacts with quantization is crucial for practical applicability. SEFT is compatible with quantization, since it focuses on the seleting weight updating while leaving the choice of numerical precision.
>
> To evaluate this compatibility, we applied 4-bit post-training quantization to SEFT-fine-tuned LLaMA2-7B and LLaMA2-13B models. As shown in Table 11, qSEFT consistently outperforms both the original 4-bit models without fine-tuning and the qLoRA baselines. These results indicate that SEFT remains effective under low-precision settings, indicating that SEFT is a promising component in practical.
>
> Moreover, we fully agree that further exploring this **hybrid compression strategy**, which combines sparsity and quantization, is a promising direction for future work, as it may unlock additional real-world gains in both memory and computational efficiency.
>
> Table 12. Performance evaluation of LLaMA2-7B and LLaMA2-13B models on MMLU under 4-bit quantization. Higher values indicate better performance.
> |model|LLaMA2-7B| LLaMA2-13B
> |-|-|-|
> |Original(4-bit)| 44.8| 54.7|
> |qLoRA| 47.7| 55.0|
> |qSEFT|**48.8**|**55.5**|
>
> > Minor • Typo at L885: “The results, presented in Table 7, show that …”.
>
> A: We thank the reviewer for pointing out this typo. We have corrected it in the revised version.

---

### Official Review · Reviewer_wBjB · 2025-10-27

**Soundness:** 3
**Presentation:** 3
**Contribution:** 2
**Rating:** 4
**Confidence:** 4

**Summary:**

The authors experiment on LLaMA, DeepSeek, and Mistral model families, pruned with both SparseGPT and Wanda. The results demonstrate that SEFT outperforms baselines like LoRA* (prune-after-tuning), SPP, and SQFT across various benchmarks (Commonsense Reasoning, MMLU, GSM8K), with a particularly strong advantage at high sparsity levels (e.g., 70%-80%). Furthermore, SEFT shows significant efficiency gains, including up to 2x memory savings during training and up to 2.5x inference speedup on a CPU.

**Strengths:**

SEFT cleverly adapts the idea of Dynamic Sparse Training (DST) to the novel scenario of "fine-tuning an already-sparse model." The "Drop-Grow-Adapt" three-step cycle shown in Figure 2 is clear and logically sound. The "Grow" step, in particular, which allows the model to "resurrect" pruned weights, is key to its task-adaptive capability.

he analyses in the appendix are thorough. For example, Appendix E.1 (Table 7) proves the necessity of "allowing resurrection of pruned weights," and Appendix E.2 (Fig 5a) demonstrates the superiority of the "sensitivity-based" pruning criterion over "magnitude-based." This greatly enhances the credibility of the method's design.

**Weaknesses:**

SEFT introduces new hyperparameters that require tuning, most notably the sparse topology "update frequency $k$" and the "drop rate." The appendix (F.3, F.4) shows that the optimal update frequency $k$ is task-dependent (e.g., 60 for C4 vs. 10 for Commonsense Reasoning). This slightly weakens the claim of a universally simple method, as this tuning adds back some of the experimental burden that fine-tuning aims to reduce.


The "Sparsity Adaptation" step (Sec 3.2) seems to re-prune the entire model at every $k$ steps using a sensitivity-based score ($|\theta_i \nabla_{\theta_i} L|$). Is this score computed on the fly, or are gradients accumulated? This step seems computationally intensive, and its cost relative to the "Drop" and "Grow" steps is not fully clarified.

**Questions:**

Could the authors please quantify the training wall-clock time of SEFT compared to the baselines (LoRA*, SPP, SQFT)? How significant is the computational overhead from the "Grow" and "Adapt" steps, which require dense gradient information?

In the "Sparsity Adaptation" step (Sec 3.2), is the sensitivity score $s_i = |\theta_i \nabla_{\theta_i} L|$ computed using the instantaneous gradient from the current batch, or is it an accumulated value (e.g., from an optimizer like Adam)? If it's instantaneous, how stable is this metric? If it's accumulated, what is the memory cost?

---

> ### Author Response · Authors · 2025-11-21
>
> Thank you very much for your detailed comments and thoughtful questions. We provide our response to your constructive comments below.
>
> > SEFT brings in some new settings you’ll need to tweak, like how often you update the sparse structure (called the update frequency $k$) and how many connections to drop (the drop rate). The appendix shows that the best value for $k$ depends on what you’re working on—for example, 60 for C4 but only 10 for Commonsense Reasoning. Because of this, the method isn’t quite as “set-and-forget” as it first seems, since you’ll still need to spend some time finding the right settings for your task.
>
> A: Thank you for your thoughtful question and concern. We conducted the sensitivity analyses on C4 and Commonsense Reasoning primarily **to understand how SEFT behaves under different training regimes**. We acknowledge that this may introduce a small amount of additional computation beforehand; however, our findings provide practical and foundational guidance for applying SEFT universally to varied fine-tuning scenarios.
>
> The main reason for testing on both C4 and Commonsense Reasoning is their differences in dataset scale and required training steps. The C4 dataset is substantially larger, leading to much more training steps, whereas Commonsense Reasoning contains far fewer samples and therefore involves fewer training steps. We explored how SEFT’s performance is affected under these distinct conditions, which **can provide practical guidance for extending SEFT to other downstream tasks**.
>
> Our findings show that the drop rate has minimal impact on performance and **can generalize across different tasks**, even when the data scales differ substantially. In contrast, the update frequency naturally correlates with the total number of training steps—longer training benefits from a longer topology-update interval, whereas shorter training requires a shorter topology-update interval to ensure adequate topology adaptation. **This principle transfers consistently across models and datasets and was directly applied to subsequent fine-tuning experiments**, such as on GSM8K, using both the DeepSeek and Mistral models, without any additional tuning.
>
> Overall, these analyses are not intended to complicate the method but rather to **demonstrate SEFT’s scalability and adaptability across diverse training scenarios**. From these experiments, our findings show that its hyperparameters behave predictably and transfer effectively to new tasks.
>
>
> > In the “Sparsity Adaptation” step (Section 3.2), it looks like the whole model gets re-pruned every $k$ steps using a score that measures how sensitive each weight is. Is this score calculated right away each time, or do you add up the gradients over several steps? This part seems like it could take a lot of computing power, but it’s not totally clear how much work it adds compared to the “Drop” and “Grow” steps.
>
> A: Thank you for your concern and for helping us clarify this point. In practice, SEFT employs **accumulated gradient** estimates rather than instantaneous gradients to **ensure greater stability** during fine-tuning, as described in line 251 of the paper, specifically, using a 5-step gradient accumulation before each update. What should be made clearer is that SEFT re-prunes only from the activated weights in the sparse base model, every $k$ steps using a sensitivity-based score.
>
> Regarding the empirical computational cost (as shown in Table 8), the sparsity adaptation step accounts for approximately 0.8 seconds out of the total 4.5 seconds required for each topology update. Importantly, because SEFT updates the topology only at regular intervals, this overhead is amortized across multiple training steps and represents only a small fraction of the overall training time. **Further discussion is provided in the next question.**
>
> Table 8. Running time (in seconds) (**↓**) for the evolution and adaptation steps using LLaMA2-7B and LLaMA3-8B on the MMLU downstream task.
>
> |model|Evolution| Adaptation| Total
> |-|-|-|-|
> |LLaMA2-7B| 3.7|0.8| 4.5
> |LLaMA3-8B| 4.3|0.9| 5.2

---

> ### Author Response · Authors · 2025-11-21
>
> > Could you share how long it actually takes to train SEFT compared to the other methods, like LoRA*, SPP, or SQFT? I’m also curious—do the “Grow” and “Adapt” steps (which need a lot of gradient info) slow things down a lot, or is the extra time pretty minor?
>
> A: Thank you for raising this important question. We report the training wall-clock time of SEFT compared with the baselines (LoRA*, SPP, SQFT) **in Table 9**, under the same training configuration (batch size, sequence length, and hardware setup).
>
> Regarding the computational overhead, **SEFT updates the topology only at regular intervals**, which keeps the amortized cost low. **As shown in Table 9** below, taking models of different sizes fine-tuned on the MMLU downstream task as an example—where the sparse topology is updated every 60 steps—the overhead introduced by **the Grow and Adapt steps accounts for only a negligible fraction of the average per-step training time**. Consequently, SEFT still achieves a faster average per-step training time than other fine-tuning baselines.
>
> Another point worth clarifying is that SEFT’s evolution procedure could, in principle, involve the entire model. We acknowledge that such full-model re-evaluation would be computationally and memory intensive; therefore, in practice, we adopt a more efficient implementation. SEFT performs grow within a candidate pool whose size equals the number of update weights $\phi$
> (approximately 1% of the base model parameters). This design substantially reduces both the computation and memory overhead required to store and use gradient information compared with full-model regrowth. The resulting total memory consumption is illustrated in Figure 3 of the main paper.
>
> These results demonstrate that SEFT not only improves memory efficiency but also reduces overall training time compared with these methods.
>
> Table 9. Average per-step running time (in seconds) (**↓**) for LLaMA2-7B and LLaMA3-8B.
> |model|LLaMA2-7B| LLaMA3-8B
> |-|-|-|
> |LoRA| 20.7| 21.6|
> |SPP| 55.6| 58.9|
> |SQFT| 38.5| 40.2|
> |SEFT wo/ Evolution| 19.2| 19.8|
> |SEFT w/ Evolution| 19.6| 20.2|
>
>
>
> > During the “Sparsity Adaptation” step (Section 3.2), do you calculate the sensitivity score for each weight using just the current batch, or do you keep track of these scores over time (like Adam does)? If it’s just for one batch, is the score pretty stable? And if you’re tracking it across batches, how much extra memory does that take?
>
> A: Thank you for your insightful question. In practice, SEFT employs **accumulated gradient** estimates rather than instantaneous gradients to ensure greater stability during fine-tuning, as described in line 251 of the paper. We totally agree that using accumulated gradients smooths out short-term fluctuations from individual batches and provides a more reliable sensitivity measure for topology adaptation.
>
> We acknowledge that this accumulated approach introduces a slight additional cost, but it remains minimal in practice. In SEFT, sparsity adaptation re-prunes weights **only from the activated sparse base model**. We manage memory efficiently by using a hook function that retrieves gradient information layer by layer, retaining only the gradients corresponding to the activated weights. As a result, SEFT still exhibits lower memory consumption compared with other baselines, **as shown in the below Table**.
>
> We have summarized the component-wise memory usage under a configuration of batch size = 1, input sequence length = 1024 in Appendix C (Figure 4(a)), where we compare SEFT with LoRA. In practice, **SEFT is more memory-efficient, especially when handling long input sequences**.
>
> Table 10. GPU memory usage (in GB) (**↓**) for SEFT and LoRA methods under different input lengths.
> |Method|Sequence length|Weight|Activation|Gradient|Optimizer|Total|
> |-|-|-|-|-|-|-|
> |SEFT|1024|14.15|13.27|1.07|1.09|**29.59**|
> |LoRA|1024|13.83|17.65|0.41|0.64|32.54|
> |SEFT|2048|14.15|39.38|1.07|1.09|**55.70**|
> |LoRA|2048|13.83|48.04|0.43|0.64|62.95|

---

### Official Review · Reviewer_rqp2 · 2025-10-31

**Soundness:** 3
**Presentation:** 4
**Contribution:** 2
**Rating:** 4
**Confidence:** 4

**Summary:**

This paper proposes Sparsity Evolution Fine-Tuning (SEFT), a new method for fine-tuning sparse large language models (LLMs). While traditional fine-tuning approaches like LoRA fail to maintain sparsity and post-training pruning methods (e.g., SparseGPT, Wanda) often cause substantial performance degradation at high sparsity, SEFT addresses these issues by dynamically evolving the sparse topology during fine-tuning while keeping the overall sparsity fixed.

**Strengths:**

- The paper is clearly presented and well-organized, with only a few minor typos.
- The proposed SEFT method is conceptually simple, well-motivated, and demonstrates strong empirical performance.
- The authors provide comprehensive ablation studies, including analyses of sparsity’s effect on performance, inference speed, and memory efficiency.
- Experiments are conducted across multiple datasets and diverse LLM families, which strengthens the empirical validation of the method.

**Weaknesses:**

- Limited Related Work Discussion:
The discussion of related work primarily focuses on low-rank methods (e.g., LoRA) but overlooks several recent sparse fine-tuning and pruning approaches that are methodologically closer to SEFT. Although Appendix A.2 and A.3 cover PEFT and dynamic sparse training, it would strengthen the paper to include recent sparse PEFT methods (e.g., [1,2,3,4]) for a more comprehensive contextualization.
- Novelty Clarification:
The paper lacks a clear explanation of how SEFT differs from similar sparse fine-tuning frameworks, particularly SpIEL [5], which also incorporate grow-and-drop phases and gradient-based parameter selection. A more explicit discussion comparing SEFT’s innovations—such as its sparsity adaptation phase or index update mechanism—with these prior works would clarify its novelty. Besides, similar to SMT[3], SEFT also introduce reconstruction indices method to maintain and train sparse model.  SEFT also have similar parameter selection apprach regards to gradient magnitude similar to SpIEL[5] and SMT[3]. More discussion about the similarity will be appreciated.
- Missing Baseline Comparisons:
SEFT is conceptually similar to several sparse fine-tuning methods (e.g., SMT [3], SpIEL [5]), yet these are not included as baselines in the experiments. Adding such comparisons would provide a more balanced and convincing evaluation of SEFT’s performance and efficiency.
- Minor Typos:
Inconsistent spelling: “fine-tuning” vs. “finetuning” → use one form (“fine-tuning”).
“enables sparse LLMs recover” → “enables sparse LLMs to recover.”
“delta vector δ dynamically explore” → “explores.”
“Specifically models are fine-tuned…” → “Specifically, models are fine-tuned…”


[1] The Lottery Ticket Hypothesis: Finding Sparse, Trainable Neural Networks

[2] Parameter-Efficient Fine-Tuning without Introducing New Latency

[3] Sparse Matrix in Large Language Model Fine-tuning

[4] Training Neural Networks with Fixed Sparse Masks

[5] Scaling Sparse Fine-Tuning to Large Language Models

**Questions:**

- Memory Cost:
SEFT maintains reconstruction indices to record sparsity patterns. Could the authors provide quantitative estimates of this additional memory overhead? Sparse matrices typically require extra storage for index data, potentially offsetting efficiency gains. Further, does maintaining and reconstructing these indices introduce additional copy operations or memory fragmentation?

- Training Speed:
Sparse matrix operations often involve index resolution and memory mapping, which may reduce cache efficiency and increase latency during the indices read operations. While the paper evaluates inference latency (main text and Appendix B), it lacks discussion of training time. Including an analysis of training speed and a comparison to dense fine-tuning methods would help clarify whether SEFT introduces measurable overhead during training.

This paper presents a promising and well-executed approach to fine-tuning sparse LLMs. However, its contribution would be strengthened by a deeper comparison to related sparse fine-tuning works, explicit clarification of novel aspects, and quantitative discussion of memory and training-time overhead.
If these concerns are addressed in the rebuttal, I would be willing to reconsider my current rating.

---

> ### Author Response · Authors · 2025-11-21
>
> We are grateful for your feedback and insightful questions. We provide point-wise responses to address your concerns below.
>
> > Limited Related Work Discussion: The discussion of related work primarily focuses on low-rank methods (e.g., LoRA) but overlooks several recent sparse fine-tuning and pruning approaches that are methodologically closer to SEFT. Although Appendix A.2 and A.3 cover PEFT and dynamic sparse training, it would strengthen the paper to include recent sparse PEFT methods (e.g., [1,2,3,4]) for a more comprehensive contextualization.
>
> A: Thank you for your valuable comment and for sharing the related papers. We fully agree and have added a discussion of these works **in Appendix A.1 (LLM Pruning) and Appendix A.2 (Parameter-Efficient Fine-Tuning), highlighted in blue in the latest revision**.
>
> For Appendix A.1 (LLM PRUNING), we add:
>
> *... However, most existing pruning methods often struggle to maintain satisfactory performance, especially under high sparsity levels. Moreover, state-of-the-art LLM pruning techniques heavily rely on calibration data, which may inadvertently remove connections critical for specific downstream tasks, leading to suboptimal performance. To preserve the efficiency benefits of pruning while maintaining accuracy, we propose SEFT, a finetuning method that maintains sparsity throughout training and enables the dynamic evolution and recovery of sparse connectivity. Unlike the method [1, 4] that rely on a dense model and update only a fixed subset of parameters during training, SEFT continually repairs task-relevant connections.*
>
> For Appendix A.2 (PARAMETER-EFFICIENT FINE-TUNING), we add:
>
> *In contrast, sparse fine-tuning, which updates only a subset of the model’s original parameters with minimal architectural changes, has emerged as a practical strategy for PEFT. However, its practical memory and computational benefits often diminish at large scale. Methods such as SIFT~\citep{songsparse}, PaFi [2], SHiRA \citep{bhardwaj2024sparse} and SpIEL[5] impose sparse updates, yet still rely on dense pretrained models and result in dense models after fine-tuning. Other approaches, including SMT [3] and GPS \citep{zhang2024gradient}, improve efficiency by selecting gradients per neuron. Nevertheless, they depend on gradient-based warm-up phases. These mechanisms add overhead from mask storage, dense model dependency, or fixing weight selection in advance, making them less suitable for sparse language model fine-tuning across diverse downstream tasks. In contrast, SEFT inherits the advantages of sparsity while dynamically recovering task-relevant connectivity and maintaining an efficient sparse topology throughout training.*

---

> ### Author Response · Authors · 2025-11-21
>
> > Novelty Clarification: The paper lacks a clear explanation of how SEFT differs from similar sparse fine-tuning frameworks, particularly SpIEL [5], ..... Besides, similar to SMT[3], SEFT also introduce reconstruction indices method to maintain and train sparse model. SEFT also have similar parameter selection apprach regards to gradient magnitude similar to SpIEL[5] and SMT[3]. More discussion about the similarity will be appreciated. Adding such comparisons would provide a more balanced and convincing evaluation of SEFT’s performance and efficiency.
>
> A: We appreciate the reviewer’s insightful comments regarding the similarity between SEFT and existing sparse fine-tuning frameworks such as SpIEL [5] and SMT [3]. Below we clarify both the conceptual overlap and the key distinctions that define SEFT’s novelty. **We further elaborate on these points in Section H of the latest revised version.**
>
> **Similarity:**
> SEFT, SpIEL, and SMT are all grounded in the paradigm of sparse fine-tuning, where only a subset of the model’s original parameters is updated with minimal architectural modification. All three methods aim to achieve parameter efficiency through sparse updates, without relying on additional adapter modules.
>
> **Key Differences / Novel Contributions:**
>
> - **Maintaining sparsity throughout training**:
> Unlike SpIEL and SMT, which **rely on dense pretrained models and result in dense model after finetuning**, SEFT preserves sparsity throughout the entire training process. After finetuning, SEFT directly produces a sparse model, without requiring additional pruning that could degrade performance, thereby maintaining the computational and memory efficiency of sparsity during both training and deployment.
>
> - **Task-aware connectivity recovery**:
> The main motivation of drop-and-grow in SEFT is to recover task-relevant connections that were previously pruned, **starting from zeroed-out parameters**. In contrast, SpIEL’s drop-and-grow mechanism and SMT’s parameter-selection strategy focus primarily on updating the existing parameters, rather than sparsity structural recovery and evolution. Moreover, SMT selects the most influential sub-matrices only during an initial warm-up phase, after which the pattern remains fixed.
>
> - **Sparsity adaptation component**:
> SEFT introduces a sparsity adaptation mechanism that maintains the target sparsity throughout finetuning. In contrast, SMT and SpIEL always keep a dense model and only perform finetuning on top of it.
>
> - **Flexibility to different sparsity patterns**:
> SEFT is designed to support various sparsity formats, including unstructured sparsity, structured sparsity, and 2:4 semi-structured sparsity. This flexibility enables SEFT to adapt to different sparse architectures and hardware constraints.
>
> In summary, while SEFT shares the general sparse fine-tuning paradigm with SpIEL and SMT, it differs substantially in **training dynamics, sparsity maintenance, and task-adaptive recovery mechanisms**. These innovations make SEFT a more practically efficient approach for fine-tuning of sparse large language models across different downstream tasks.
>
> Furthermore, we compared SEFT with SMT and SpIEL under the same fine-tuning setup on the Commonsense Reasoning task for LLaMA2-7B and LLaMA3-8B on sparsity level of 60% after fine-tuning. As shown in Table 5, SEFT achieves superior performance on both model sizes.
>
> Table 5. Performance on LM-Eval (↑) for LLaMA2-7B and LLaMA3-8B.
> |model|LLaMA2-7B| LLaMA3-8B
> |-|-|-|
> |SMT| 55.4| 57.6|
> |SpIEL| 55.6| 57.9|
> |SEFT| 56.5| 58.4|
>
>
> > Minor Typos: Inconsistent spelling: “fine-tuning” vs. “finetuning” → use one form (“fine-tuning”). “enables sparse LLMs recover” → “enables sparse LLMs to recover.” “delta vector δ dynamically explore” → “explores.” “Specifically models are fine-tuned…” → “Specifically, models are fine-tuned…”
>
> A: We thank the reviewer for pointing out these typos. We have corrected all of them in the revised version.

---

> ### Author Response · Authors · 2025-11-21
>
> > Memory Cost: SEFT maintains reconstruction indices to record sparsity patterns. Could the authors provide quantitative estimates of this additional memory overhead? Sparse matrices typically require extra storage for index data, potentially offsetting efficiency gains. Further, does maintaining and reconstructing these indices introduce additional copy operations or memory fragmentation?
>
> A: Thank you for your question. Indeed, SEFT maintains reconstruction indices to record sparsity patterns. In particular, SEFT stores the update indices $\phi$ that correspond to the active parameters in the base model. This design **introduces a small additional memory cost, as shown in the “Weights” column of Table 6**. However, the overall memory efficiency of SEFT remains highly favorable due to the following factors:
>
> **1. Activation storage saving:**
> Unlike LoRA, SEFT does not introduce any additional adapter layers. All parameter updates are performed in-place within the existing model tensors. As a result, SEFT avoids the extra activation memory required to compute and store adapter outputs, leading to a measurable reduction in peak memory usage during both forward and backward passes.
>
> **2. Minimal index overhead:**
> The additional memory required to store reconstruction indices is typically very small. Each index entry corresponds to one trainable parameter and is stored in integer form, contributing only a minor portion of the total memory footprint, especially when the proportion of updated parameters is low (typically around 0.3% of the dense model during fine-tuning).
>
> We have summarized the component-wise memory usage under a configuration of batch size = 1, input sequence length = 1024 in Appendix C (Figure 4(a)), where we compare SEFT with LoRA. A detailed comparison of memory usage is also provided in Table 6 below. In practice, **SEFT is more memory-efficient than LoRA, especially when handling long input sequences**.
>
>
> Table 6. GPU memory usage (in GB) (**↓**) for SEFT and LoRA methods under different input lengths.
> |Method|Sequence length|Weight|Activation|Gradient|Optimizer|Total|
> |-|-|-|-|-|-|-|
> |SEFT|1024|14.15|13.27|1.07|1.09|**29.59**|
> |LoRA|1024|13.83|17.65|0.41|0.64|32.54|
> |SEFT|2048|14.15|39.38|1.07|1.09|**55.70**|
> |LoRA|2048|13.83|48.04|0.43|0.64|62.95|
>
> > Training Speed: Sparse matrix operations often involve index resolution and memory mapping, which may reduce cache efficiency and increase latency during the indices read operations. While the paper evaluates inference latency (main text and Appendix B), it lacks discussion of training time.
>
> A: Thank you for raising this important point, and we appreciate the opportunity to clarify SEFT’s training efficiency. While sparse matrix operations can, in principle, introduce additional latency, our empirical results show that SEFT introduces only a modest overhead for index updates during training. We adopt an efficient strategy that triggers **topology updates only at regular intervals, keeping the amortized cost per update minimal relative to the overall training time**.
>
> As shown in the Table 7 below, across different model sizes fine-tuned on the MMLU downstream task, the amortized overhead of index evolution is approximately **0.4 seconds per step throughout the training process**. SEFT also achieves a faster average per-step training time than other fine-tuning baselines. This demonstrates that SEFT **not only improves memory efficiency but also reduces overall training time** compared with existing methods.
>
> Table 7. Average per-step running time (in seconds) (**↓**) for LLaMA2-7B and LLaMA3-8B.
> |model|LLaMA2-7B| LLaMA3-8B
> |-|-|-|
> |LoRA| 20.7| 21.6|
> |SPP| 55.6| 58.9|
> |SQFT| 38.5| 40.2|
> |SEFT wo/ Evolution| 19.2| 19.8|
> |SEFT w/ Evolution| 19.6| 20.2|
>
> We hope our explanations and proposed revisions address your concerns, and we would greatly appreciate it if you could reconsider your score. Thanks again for your time and efforts!

---

> ### Author Response · Authors · 2025-11-27
>
> Dear Reviewer rqp2,
>
> We thank you for your constructive feedback on our work and for taking the time to review it. We hope our answers and results above have addressed your concerns. If so, we would greatly appreciate it if you could reconsider your score. If you have any additional questions or would like further clarification. We are willing to engage in further discussions. Thanks again for your time and effort.

---

### Official Review · Reviewer_Jmj9 · 2025-11-02

**Soundness:** 2
**Presentation:** 2
**Contribution:** 2
**Rating:** 6
**Confidence:** 3

**Summary:**

This paper proposes Sparsity Evolution Fine-Tuning (SEFT), a sparse fine-tuning method for large language models (LLMs) that dynamically adjusts the sparsity topology during training. SEFT employs a drop-and-grow mechanism to evolve sparse connections, removing weights with the smallest update magnitudes and reactivating previously pruned ones with the largest current gradients. This allows the sparsity pattern to adapt throughout fine-tuning while maintaining a fixed sparsity ratio. Experiments on models such as Llama and Mistral demonstrate that SEFT achieves better performance and higher efficiency than existing sparse fine-tuning baselines.

**Strengths:**

1. Efficient fine-tuning for large-scale LLMs is a pressing and valuable research direction. Leveraging sparsity to reduce training costs offers clear benefits for resource-constrained settings.
2. The motivation is sound and design is reasonable. Allowing the model to reconstruct its sparse topology and reactivate pruned weights provides flexibility for adapting to downstream tasks, addressing the rigidity of static pruning schemes.
3. Comprehensive experiments and evaluation settings. The experiments cover multiple open-source LLM architectures and two different pruning schemes as initialization. The analysis includes both performance and efficiency metrics, with additional ablations and sensitivity studies in the appendix, lending credibility to the conclusions.
4. Clear writing and presentation. The paper is well structured, and Figure 1 offers an intuitive comparison between SEFT and LoRA, clearly illustrating the mechanism.

**Weaknesses:**

1. Missing comparisons to recent baselines. The paper omits comparisons with recent methods such as SparseLoRA [1] and S^$2$FT [2], which are directly relevant. Even a conceptual or mechanism-level discussion would help position SEFT more clearly within the current literature.
2. Lack of statistical significance analysis. While SEFT often achieves the best average results (e.g., Table 2), the margins are small (~1%). Given the stochastic nature of LLM fine-tuning, reporting averages and standard deviations over multiple seeds or conducting significance tests would strengthen claims such as “SEFT consistently outperforms baselines”.
3. Insufficient discussion of computational overhead of “drop-and-grow”. The drop-and-grow updates rely on dense gradient computation (Appendix I), yet the main text lacks quantitative analysis of this cost. Although the proposed update scheme mitigates overhead, it may still limit SEFT’s deployability in low-resource environments.

---

**References**

[1] Khaki, S., Li, X., Guo, J., Zhu, L., Plataniotis, K.N., Yazdanbakhsh, A., Keutzer, K., Han, S. &amp; Liu, Z.. (2025). SparseLoRA: Accelerating LLM Fine-Tuning with Contextual Sparsity. Proceedings of the 42nd International Conference on Machine Learning, in Proceedings of Machine Learning Research 267:29768-29783.

[2] Yang, X., Leng, J., Guo, G., Zhao, J., Nakada, R., Zhang, L., ... & Chen, B. (2024). S $^{2} $ FT: Efficient, scalable and generalizable LLM fine-tuning by structured sparsity. Advances in Neural Information Processing Systems, 37, 59912-59947.

**Questions:**

Could the authors quantify the time cost of the topology evolution step relative to the total training process? Would accounting for this overhead alter the comparative efficiency results between SEFT and other methods?

---

> ### Author Response · Authors · 2025-11-21
>
> Thank you very much for your detailed comments and thoughtful questions.
>
> > w1: Missing comparisons to recent baselines. The paper omits comparisons with recent methods such as SparseLoRA [1] and S^FT [2], which are directly relevant. Even a conceptual or mechanism-level discussion would help position SEFT more clearly within the current literature.
>
> A: Thank you for this valuable comment. We appreciate the reviewer’s suggestion to discuss recent fine-tuning methods such as SparseLoRA [1] and S^2FT [2].
>
> **Conceptual Comparison:**
>
> 1. SparseLoRA is still a LoRA-based finetuning paradigm. In contrast, SEFT introduces a new parameter-efficient paradigm: instead of using low-rank adapters, we directly select and finetune the most effective parameters. This makes SEFT conceptually different from LoRA.
>
> 2. The reason we adopt this new fine-tuning paradigm is that it is naturally **tailored for sparse models**, aiming to update the sparse topology according to the downstream task while maintaining sparsity efficiency. In contrast, SparseLoRA produces a dense model during fine-tuning, and requires an additional pruning stage afterward to achieve efficiency, often resulting in degraded final performance.
>
>
> 3. S^2FT structually selects a subset of weights for finetuning but still need to maintains a **dense** model. In contrast, SEFT starts from a sparse model and dynamically evolves its sparsity topology to adapt to downstream tasks, preserving sparsity throughout training and ultimately producing a high-performing **sparse** model.
>
> **Empirical Comparison:**
>
> We additionally compared SEFT with SparseLoRA and S^2FT under the same fine-tuning setup on the Commonsense Reasoning task for LLaMA2-7B and LLaMA3-8B with sparsity level of 60%. **As shown in Table 1**, SEFT achieves superior performance on both model sizes.
>
> We have added a detailed discussion **in Section A.2** of the revised manuscript to clarify SEFT’s role and advantages within the current sparse fine-tuning literature.
>
>
> Table 1. Performance comparison on LM-Eval (↑) for LLaMA2-7B and LLaMA3-8B.
> |Model|LLaMA2-7B | LLaMA3-8B
> |-|-|-|
> |SparseLoRA| 55.5| 57.2|
> |S^2FT| 54.7| 57.8|
> |SEFT| 56.5| 58.4
>
>
> > Lack of statistical significance analysis. While SEFT often achieves the best average results (e.g., Table 2), the margins are small (~1%). Given the stochastic nature of LLM fine-tuning, reporting averages and standard deviations over multiple seeds or conducting significance tests would strengthen claims such as “SEFT consistently outperforms baselines”.
>
> A: We thank the reviewer for this insightful comment regarding statistical significance analysis. To evaluate SEFT’s performance, our experiments **span multiple models and multiple tasks**. Specifically, SEFT is tested across diverse architectures (LLaMA, DeepSeek, and Mistral) and **model sizes, as well as on a broad suite of downstream tasks**. The improvements observed across these heterogeneous settings show that SEFT’s advantages reflect a reliable performance trend.
>
> Furthermore, in our preliminary experiments, we observed that the variance in LLM fine-tuning is relatively small in our setting. However, performing multi-seed fine-tuning for large-scale models (e.g., LLaMA, DeepSeek, and Mistral) would be extremely computationally expensive. Given these constraints, we prioritize comprehensive cross-model and cross-task evaluation, which offers a more practical and meaningful assessment of SEFT’s generalization and stability.

---

> ### Author Response · Authors · 2025-11-21
>
> > Insufficient discussion of computational overhead of “drop-and-grow”. The drop-and-grow updates rely on dense gradient computation (Appendix I), yet the main text lacks quantitative analysis of this cost. Although the proposed update scheme mitigates overhead, it may still limit SEFT’s deployability in low-resource environments.
>
> A: Thank you for this valuable comment. We acknowledge that SEFT’s sparse **topology evolution with drop-and-grow** relies on dense gradient computation, which does introduce additional cost. However, in practice, we mitigate this overhead **through an efficient update scheme**. Instead of evolving the topology at every step, SEFT updates the sparsity pattern **only at regular intervals**, ensuring that the extra computation remains minimal.
>
> As shown in Table 3, taking the MMLU downstream task as an example, the topology is updated every 60 steps with 5-step gradient accumulation performed beforehand, each of which adds approximately 4.5 seconds. When amortized over the 60-step interval, the drop-and-grow phase **contributes only a negligible fraction of the overall training time.** In addition, as discussed in Section F.4 and Figure 6, using a reasonable update frequency **does not significantly affect performance**.
>
> Furthermore, we recognize that integrating sparse-aware backpropagation techniques, could further eliminate the need for dense gradient computation. This represents a promising direction for future research to address the limitations of current deep-learning frameworks and fully unlock the efficiency of sparse training in low-resource environments.
>
> Table 3. Average Running time (in seconds) (**↓**) for the evolution step, and the average per-step running time on LLaMA2-7B and LLaMA3-8B.
> |Model|Evolution| wo/ Evolution (mean)| w/ Evolution (mean)
> |-|-|-|-|
> |LLaMA2-7B| 0.4 | 19.2| 19.6
> |LLaMA3-8B| 0.4 | 19.8| 20.2
>
> > Could the authors quantify the time cost of the topology evolution step relative to the total training process? Would accounting for this overhead alter the comparative efficiency results between SEFT and other methods?
>
> A: Thank you for the question. Indeed, the topology evolution step with drop-and-grow introduces some additional cost. However, in practice, these updates are triggered only **at regular intervals**, and each update adds only a negligible amount of extra computation to the overall training time. As shown in Table 4, across different model sizes fine-tuned on the MMLU downstream task, the amortized **overhead of topology evolution is approximately 0.4 seconds per step** throughout the training process.
>
> Importantly, even after accounting for this overhead, SEFT remains faster than all baselines, as reflected in the average per-step runtime comparisons shown below. Therefore, incorporating **the cost of topology evolution does not alter SEFT’s comparative efficiency advantage over other methods**.
>
> Table 4. Comparison of average per-step running time (in seconds) (**↓**) for LLaMA2-7B and LLaMA3-8B.
> |model|LLaMA2-7B| LLaMA3-8B
> |-|-|-|
> |LoRA| 20.7| 21.6|
> |SPP| 55.6| 58.9|
> |SQFT| 38.5| 40.2|
> |SEFT wo/ Evolution| 19.2| 19.8|
> |SEFT w/ Evolution| 19.6| 20.2|

---

### Author Response · Authors · 2025-11-21

Dear Area Chair,

Thank you for your time and effort in handling our submission. We would also like to clarify one point regarding the review process. **Reviewer wBjB continued to update comments until November 17th and increased the score from 4 to 6 on that date, prior to our rebuttal submission**, as recorded in the system (see [Link](https://openreview.net/revisions?id=qFsRJSN37y)). Based on this timeline, we believe the effective score is **(6, 6, 6, 4)**. We mention this solely to ensure an fair reflection of the review timeline. Although **Reviewer rqp2 gave a score of 4 and haven't participated in the discussion, and indicated a willingness to increase the score**.

We are delighted that the reviewers recognized:

- **The clarity of our presentation and the well-organized structure** [Reviewers Jmj9, rqp2, wBjB, dRk7].
- **The thoughtful experiments**, along with the comprehensive ablation and sensitivity analyses [Reviewers Jmj9, rqp2].
- The **motivation and design are sound and reasonable** [Reviewer Jmj9, dRk7], **conceptually simple** [Reviewer rqp2], and **offering new insights** [Reviewer wBjB].
- The benefit of **reducing training costs on resource-constrained settings** [Reviewer Jmj9] and **strong empirical performance** [Reviewer rqp2].

Below is a brief summary of the reviewers’ concerns and our clarifications.

## Reviewer Jmj9
### rating: 6, no response yet
Reviewer Jmj9’s primary concerns relate to the  (i) additional baselines,  (ii) statistical significance analysis,  (iii) the computational overhead of the ‘drop-and-grow', and  (iv) the efficiency comparison with other methods.

In response,

(i) We have clarified the distinctions in **Appendix A.2** and added empirical comparisons in **Table 1**, showing that SEFT outperforms the added baselines on both LLaMA2-7B and LLaMA3-8B.

(ii) We have also clarified the SEFT’s generalization and stability **cross-models and cross-tasks**.

(iii) We reported the average running time of the “drop-and-grow”, which constitutes only **about 2%** (0.4/19.6) of the total computational cost (Table 3).

(iv) In addition, we included per-step runtime comparisons with other methods in **Table 4**, demonstrating SEFT’s clear efficiency advantages.

## Reviewer rqp2
### rating: 4, no response yet but indicated willingness to increase the score

Reviewer rqp2’s main concerns relate to (i) the limited related-work discussion and clarity of novelty, (ii) comparisons with other baselines (iii) memory cost and training speed.

In response:

(i) We have expanded the related-work discussion in **Appendix A.1 and Appendix A.2**, and clarified the key differences and the novelty of our approach in **Appendix H**.

(ii) We have added empirical comparisons with other baselines, showing that our method achieves superior performance (**Table 5**).

(iii) We have included detailed GPU memory usage in **Table 6** and reported running time comparisons in **Table 7**, demonstrating that our method **not only is more memory-efficient but also reduces overall training time**.

**With these clarifications and newly added results, we believe our rebuttal fully addresses Reviewer rqp2’s concerns.**

## Reviewer wBjB
### rating: 4 to 6 on November 17th, before we posted the rebuttal.

Reviewer wBjB’s main concerns relate to (i) hyperparameter selection (ii) the computation of "Sparsity Adaptation", (iii) momeory cost, and (iv) training time compared to the baselines.

In response:

(i) We have clarified that our hyperparameter choices are robust and transfer effectively across tasks.

(ii) We have shown that the computation of the Sparsity Adaptation step is relatively small in the overall training process, **adding only about 0.2s per step, compared to an average step time exceeding 19s**, because it is performed intermittently rather than at every step (Tables 8 and 9).

(iii) We have also shown that SEFT **does not incur high memory overhead and is more memory-efficient** than the baselines, as in Table 10.

(iv) Our new results demonstrate that SEFT is substantially **faster than the baselines (Table 9)**, showing improvements not only in memory efficiency but also in overall training time.

## Reviewer dRk7
### rating: 6, no response yet

Reviewer dRk7’s main concerns relate to (i) training time of SEFT compared to the baselines, (ii) combination with quantization.

In response:

(i) We have provided average per-step running time comparisons, demonstrating that our method **reduces overall training time** relative to existing approaches (Table 11).

(ii) We have included empirical results for our SEFT combined with quantization, showing that our method **is compatible with quantization**, outperforms qLoRA, and achieves better performance than using quantization alone. (Table 12).

We sincerely appreciate your time and thoughtful evaluation. We hope this brief summary and the additional clarifications help clarify our work and support your evaluation process.

---

### Meta-Review · Area_Chair_kX4S · 2026-01-05

**Summary:**

This paper presents a fine-tuning algorithm for LLMs with sparse weights, where the adapter parameters are updated in a way that preserves the sparsity structure. A heuristic-based algorithm is proposed to iteratively adjust both the sparsity locations ("drop" and "grow" phases) and the non-zero weight values (the "adapt" phase). The sparsity adjustment is based on magnitude of gradient updates (top-k), and is only executed periodically to trade-off efficiency due to the computation on the full weight matrices.

The reviewers raised questions regarding missing baselines, unclear novelty compared with existing literature, significance of the empirical gains, and overhead from dense gradients. The authors have provided detailed response addressing many of the initial concerns.

Overall, the proposed solution is reasonable and shows practical promises. On the down side, the training algorithm is mainly based on intuitive heuristics (just gradient magnitude) and does come with the tradeoff of either computation overhead or extra tuning efforts (e.g., "every k steps"). The empirical gains could be more convincing:
* Relative gains: in many cases, the accuracy improvements over baselines are marginal (e.g., Table 1 shows <1% gains; also consider that the proposed model introduces many additional heuristics to tune).
* Absolute performance: for both baselines and proposed models, the accuracy on many well-established benchmarks (e.g., Tables 2 & 3) are much lower than the dense models (not shown in the paper, but judging from datapoints in the literature; e.g., LLaMA 3 8B on ARC-c should be able to reach >80%, compared with ~40% shown in the paper), leaving the practical significance of the sparse tuning unclear. I suggest the authors include the performance of base models (with / without LoRA SFT) to get a clearer picture.
* (non critical as this is not brought up by reviewers) Results are mainly on the LLaMA family (and out-dated versions). It'd strengthen the empirical evidence to evaluate on other & more recent model families.

Thus, although this paper has its merits, I still think that it will be stronger by including either more theoretical understanding or more convincing empirical results.

**Reviewer Concerns:**

## Reviewer Jmj9

**Comparison with more baselines**: partially addressed by rebuttal. The authors provided additional results on a single benchmark LM-Eval, showing moderate performance gains.

**Statistical significance**: partially addressed. The authors provide rationale for sticking to single-seeded experiments due to resource constraints, which is reasonable. However, I agree with the reviewer that the margins of the gains are small and the SFT task performance is stochastic by nature.

**Computation overhead**: partially addressed by the authors' clarification and additional empirical evidence. While the runtime overhead can be amortized by periodically calling "drop-and-grow", it introduces additional factor in tuning the flexibility-complexity tradeoff, especially considering that the overall SFT training steps are often short.


## Reviewer rqp2

**Limited related work and clarification on novelty**: addressed by the rebuttal.

**Missing baselines**: partially addressed by the rebuttal by results on the single benchmark (LM-Eval)

**Memory and computation overhead**: addressed by the rebuttal through conceptual explanation and empirical measurements. However, the additional measurements on LoRA only include memory and runtime, but not accuracy (also, in the paper, there are only results on LoRA*, but not LoRA). Thus, the overall comparison with regular LoRA is not entirely clear.


## Reviewer wBjB

**Additional hyperparameters to tune**: clarified during rebuttal. The additional tuning efforts are the (probably not so critical) drawbacks of the model

**Overhead in "drop-and-grow" step**: the authors clarified the setup for gradient accumulation, and clarified the additional heuristic (candidate pool) for addressing the intensive full-parameter update. This is an additional practical "trick" that lacks in-depth understanding, making the full pipeline engineering heavy.

**Runtime compared with LoRA**: see above for lack of accuracy of "regular LoRA".

## Reviewer dRk7

**Overhead from full gradient update**: See comment above

**Extension on quantization**: addressed with additional experiments

**Reviewer Scores:**

Some reviewers may moderately increase the score after the rebuttal.

---

### Decision · Program_Chairs · 2026-01-26

Reject